# Efficient molecular evolution to generate enantioselective enzymes using a dual-channel microfluidic droplet screening platform

Fuqiang Ma[1], Meng Ting Chung[2], Yuan Yao[3], Robert Nidetz[2], Lap Man Lee[2], Allen P. Liu[2,4], Yan Feng[1], Katsuo Kurabayashi[2,5] & Guang-Yu Yang[1]

Directed evolution has long been a key strategy to generate enzymes with desired properties like high selectivity, but experimental barriers and analytical costs of screening enormous mutant libraries have limited such efforts. Here, we describe an ultrahigh-throughput dual-channel microfluidic droplet screening system that can be used to screen up to ~$10^7$ enzyme variants per day. As an example case, we use the system to engineer the enantioselectivity of an esterase to preferentially produce desired enantiomers of profens, an important class of anti-inflammatory drugs. Using two types of screening working modes over the course of five rounds of directed evolution, we identify (from among 5 million mutants) a variant with 700-fold improved enantioselectivity for the desired (*S*)-profens. We thus demonstrate that this screening platform can be used to rapidly generate enzymes with desired enzymatic properties like enantiospecificity, chemospecificity, and regiospecificity.

[1] State Key Laboratory of Microbial Metabolism, School of Life Sciences and Biotechnology, Shanghai Jiao Tong University, Shanghai 200240, China. [2] Department of Mechanical Engineering, University of Michigan, Ann Arbor, MI 48109, USA. [3] MIIT Key Laboratory of Critical Materials Technology for New Energy Conversion and Storage, School of Chemistry and Chemical Engineering, Harbin Institute of Technology, Harbin 150001, China. [4] Department of Biomedical Engineering, University of Michigan, Ann Arbor, MI 48109, USA. [5] Department of Electrical Engineering and Computer Science, University of Michigan, Ann Arbor, MI 48109, USA. These authors contributed equally: Fuqiang Ma, Meng Ting Chung. Correspondence and requests for materials should be addressed to K.K. (email: katsuo@umich.edu) or to G.-Y.Y. (email: yanggy@sjtu.edu.cn)

The highly efficient and environmentally friendly nature of enzymes has long suggested these proteins as promising alternatives to traditional inorganic catalysts. Much of our ability to synthesize chiral chemicals using enzymes relies heavily on their reaction selectivity to produce the desired enantiomeric form of desired product. Unfortunately, enzymes that exhibit high enantioselectivity are quite scarce in nature, which has severely limited the use of proteins as catalysts in practical synthetic applications (this is also true for other properties like regioselectivity and chemoselectivity). Nevertheless, directed evolution has been developed as a powerful technique for enhancing various properties of natural enzymes, typically by means of iterative rounds of mutagenesis followed by high-throughput screening[1–6].

The success of directed evolution approaches has been heavily determined by whether a scientist can find the rare hits that exhibit improved functionality from a tremendously large number of mutants. Screening such huge mutagenesis libraries for the directed evolution of enzymatic enantioselectivity has been especially challenging, owing to the requirement of performing chiral chromatography, a high-cost and labor-intensive analytical process. Previous attempts to improve enzymatic enantioselectivity have been carried out in microtiter plates, where two parallel assays on the same mutant are performed to evaluate a given mutant's enzymatic activities toward both enantiomers[7–9]. The throughput of such methods has been limited to fewer than $10^4$ colonies per day, which is only a tiny proportion of a typical mutagenesis library[10]. Ultrahigh-throughput droplet-based microfluidic techniques have shown great promise to significantly advance screening capabilities[11–15]. However, existing droplet technologies lose some of this promise when one considers that they lack the ability to screen for enzymatic enantioselectivity. This prohibits further revolutionary improvements in directed evolution-driven enzyme engineering.

Here, we report a significantly enhanced ability to screen directed evolution libraries for improvements in enzymatic enantioselectivity using a technique called "dual-channel microfluidic droplet screening (DMDS)." This ultrahigh-throughput screening method based on a lab-on-a-chip technology enables us to evaluate more than $>10^8$ droplets ($\sim10^7$ enzyme variants) per day. Our system is capable of evaluating two reaction channels simultaneously, through the use of a dual-fluorescence detection/sorting microfluidic device. The use of different combinations of two-color fluorogenic substrates allows us to screen for enzyme variants that have both improved catalytic activity and improvement for an additional enzymatic property such as regioselectivity, chemoselectivity, and enantioselectivity. As a model, we use our DMDS platform to improve the enantioselectivity of an esterase from *Archaeoglobus fulgidus* (AFEST) toward (*S*)-profens. Beyond demonstrating the rapid generation of an AFEST variant with a more than 700-fold improvement in enantioselectivity, our study underscores that the DMDS platform can be used generally in directed evolution applications for the engineering and improvement of a wide spectrum of enzymes.

## Results

**Screening of enzyme selectivity with the DMDS platform.** The DMDS platform used in our study enables the measurement two enzymatic reactions in parallel to directly evaluate enzyme selectivity (i.e., catalytic activity per se and additional properties such as regioselectivity, chemoselectivity, or enantioselectivity). Key functionalities of the system include generation of monodispersed cell-encapsulating droplets, dual-color fluorescence detection, and droplet sorting (Fig. 1a, Supplementary Fig. 1). DMDS incorporates two microfluidic units: (i) a flow-focusing droplet-generating (FFDG) device with a nozzle size of 20 μm, which is optimized based on the design reported by Hollfelder's group (Fig. 1b, c)[10,16] and (ii) a dual-channel fluorescence-activated droplet sorting (DC-FADS) device (Fig. 1d, e, also see the micro-fabrication detail in Supplementary Fig. 2). Monodispersed droplets of tunable diameter (ranging from 24 to 42 μm) can be generated by adjusting the liquid flow rates in the FFDG device (5–40 kHz) (Supplementary Fig. 3). The DC-FADS device is equipped with two sets of excitation/emission bands and a double-gated control algorithm capable of processing fluorescence signal from the same droplet (Fig. 1f).

The embedded fiber optics configuration of our DMDS DC-FADS device can efficiently suppress the background noise from a white light source and from excitation lasers. This feature enables us to conveniently operate our system at a visible environment, with a limit-of-detection as low as 10 nM for fluorescein. This sensitivity is sufficiently high to enable the discrete measurement of enzymatic activity toward both a preferred and a non-preferred enantiomer, which is an important feature for obtaining mutant variants with high selectivity across multiple rounds of directed evolution. Additionally, the spatially separated locations of the two excitation beams used with DMDS allow monitoring separate signal readouts at two different time points. Importantly, this setup maintains sufficiently low levels of crosstalk between reporter fluorophores to enable screening strategies that employ multiple fluorophores (Fig. 1f). Our DC-FADS device achieves a sorting rate of 1400 droplets s$^{-1}$ at ~100 Vp-p and with a low false-positive rate of ~3 in 10,000 (Supplementary Fig. 4). We found that the device can perform stably for at least 5 h. Thus, our system can be easily used to screen ~20 million droplets in a single experiment.

It can be perceived that the capability to evaluate enzymatic activities toward two substrates simultaneously gives the DMDS platform a significant advantage over single-substrate FADS systems.[11] Specifically, whereas it is possible that single-substrate FADS systems could in theory identify mutants that arise from the evolution of a binding site for a substrate-appended fluorophore (i.e., enzymes that do have improved activity, but only for an artificial substrate–fluorophore conjugate)[17], the DMDS platform offers the opportunity to overcome this problem by enabling the use of two substrates that are comprised of the same reactive group but have different fluorophores[18]. Thus, any mutants that evolved specificity for the fluorophore binding site would show improved activity toward only one substrate, while mutants that have evolved catalytic performance for the reactive group should show improved activity toward both substrates: we define this two selection substrate screening mode as the "DMDS cooperative mode" (sorting the double-positive population in Fig. 1g).

DMDS also enables the use of two substrates that differ in their reactive groups and differ in their fluorophores, so positive and negative selection biases can be selectively applied to isolate mutants with improved activity toward one substrate but with decreased activity toward another substrate. We define this type of screening as the "DMDS biased mode" (biased 1 and biased 2 in Fig. 1g); it is suitable for the screening of more complicated enzymatic properties such as enantioselectivity, chemoselectivity, and regioselectivity. Note that, in the DMDS biased mode, by exchanging the functional moieties and fluorophores from the original set of substrates, one can actually conduct a "secondary screening," which can ensure that identified mutants are indeed evolved for the reactive group.

**Evolution of the enantioselectivity of AFEST.** To demonstrate the robustness of our DMDS system, we chose an important class

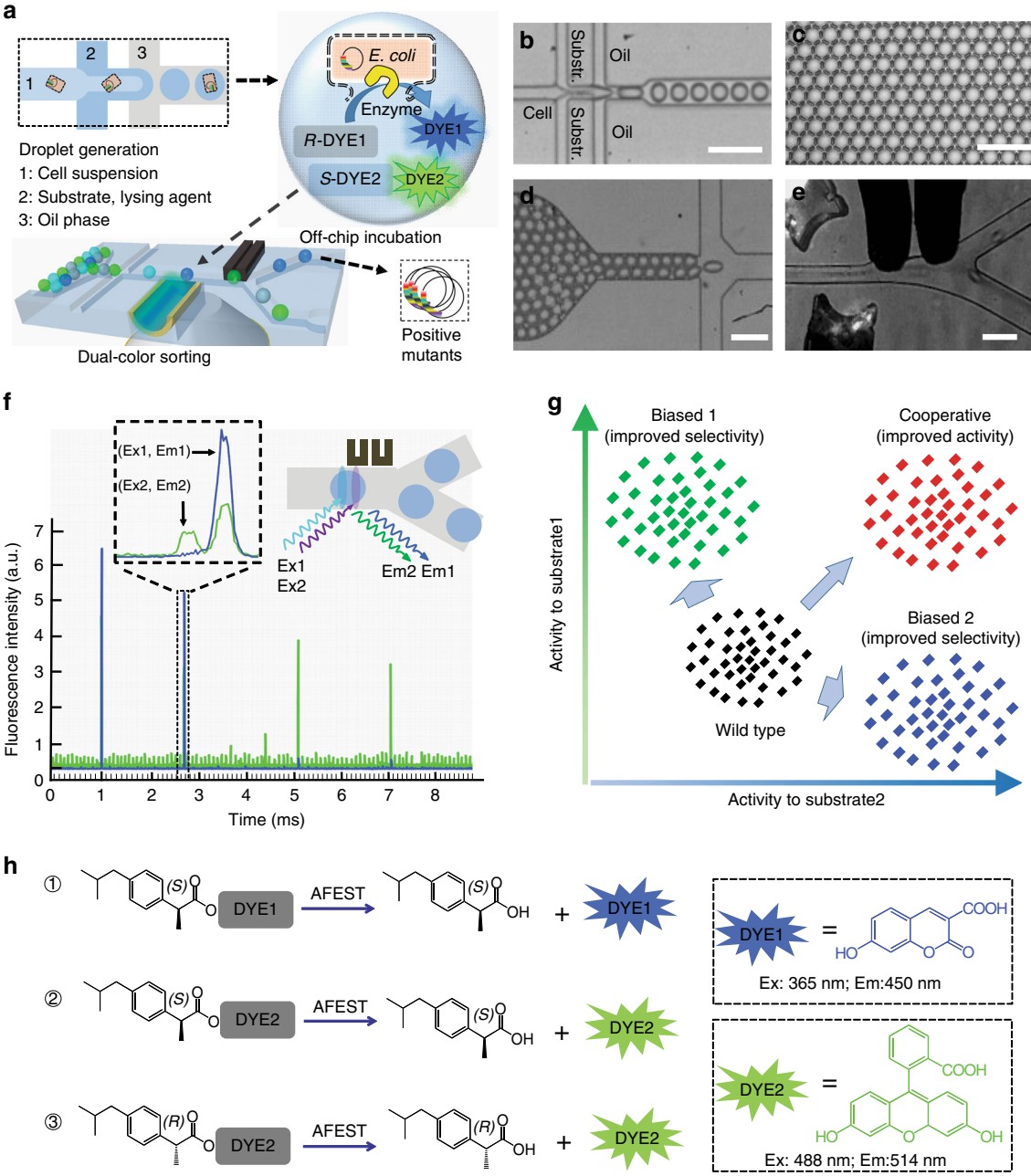

**Fig. 1** DMDS platform for screening enzymatic enantioselectivity. **a** Schematic of DMDS operation. Mutant enzyme-expressing single cells are encapsulated in water-in-oil droplets with two fluorogenic substrates and lysis buffer. After the droplets are incubated for a specified time, those droplets containing the desired mutants are enriched via fluorescence-activated droplet sorting. Optical images of DMDS processes: **b** droplet generation; **c** off-chip incubation; **d** droplet reinjection; **e** fluorescence-activated droplet sorting. **f** To avoid crosstalk of two fluorescence signals, the droplets are excited by two spatially separated lasers, which generates two temporally separated emissions. **g** Sorting different populations in a mutation library with the DMDS platform is achieved via two screening modes: a cooperative mode and biased modes. **h** Three fluorogenic substrate designs and their enzymatic reactions yielding two different fluorescence signals. Scale bars: 100 μm

of anti-inflammatory drugs, profens, as the model substrates. Profens are chiral molecules and widely used to treat pain, inflammation, fever, and stiffness. The curative effects of profen drugs come from the ($S$)-enantiomers[19], while the ($R$)-profens may cause serious side effects, by disrupting normal lipid metabolism and membrane function[20]. Therefore, it would be strongly preferable to use pure ($S$)-profens as the anti-inflammatory drug instead of currently used racemic mixtures of profens. AFEST is a thermophilic esterase from *A. fulgidus* that showed high activity toward profen esters. Its specific activity toward ibuprofen-*p*-nitrophenol (*p*NP) ester is 10 μmol min$^{-1}$ mg$^{-1}$ at 37 °C, which is

approximately 200–500-fold higher than the well-know commercialized enzymes CALA[21] and CALB[22] from *Candida antarctica*. However, the wild-type AFEST shows a slight preference toward ($R$)-profens instead of the desired ($S$)-profens. Here, we sought to use the DMDS platform to evolve AFEST to be a highly enantioselective enzyme.

To improve the enantioselectivity of AFEST, we designed and synthesized a set of fluorogenic substrates by esterifying ($S$)-ibuprofen and ($R$)-ibuprofen with different fluorophores (Fig. 1h; the synthesis details were illustrated in Supplementary Fig. 5 and described in Supplementary Methods). Two combinations of

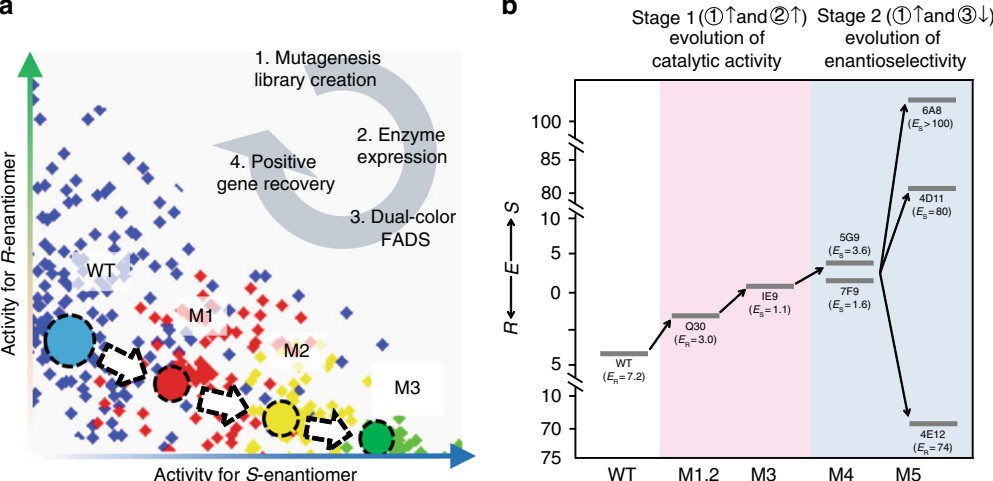

**Fig. 2** Directed evolution of the enantioselectivity of AFEST. **a** Conceptual progression of enzymatic enantioselectivity enhancement by iterative rounds of mutagenesis and use of the DMDS process. **b** Cumulative improvement in the enantioselectivity of AFEST resulting from the various directed evolution steps of the present study

different substrates were used for the two operation modes of the DMDS platform: substrate **1** and **2** were used for the cooperative mode and substrate **1** and **3** were used for the biased mode.

We performed a total of five rounds of mutagenesis and screening for *afest* directed evolution (Fig. 2a). The whole screening process is summarized in Supplementary Table 1. First, the DMDS platform was operated in the cooperative mode to improve the catalytic activity of AFEST toward (*S*)-ibuprofen. The first round of random mutagenesis used the wild-type *afest* gene as a template. Error-prone PCR was used to generate a variant library comprised of approximately 2 million mutants that harbored between 2 and 4 nucleotide mutation sites per variant (Library 1). Next, this library was subjected to an enrichment iteration of DMDS to increase the proportion of positive mutants in a condensed library: >99% of the ~400 variants in the condensed library showed activity for (*S*)-ibuprofen (Supplementary Fig. 6a). The enzymatic performance of positive variants was verified using a 96-well plate format assay. We selected seven positive mutant AFEST variants, among which there were a total of nine mutation sites (Supplementary Table 2).

The next round of directed evolution of *afest* consisted of shuffling of these nine mutation sites, generating a DNA shuffling library comprised of approximately 500,000 variants (Library 2). After one round of DMDS screening of this DNA shuffling library in cooperative mode and subsequent verification of ~200 variants in 96-well plate assays (100% of which showed activity for (*S*)-ibuprofen, Supplementary Fig. 6b), a three-site mutant, Q30 (I209V/D211G/L257P), was identified as the best performing mutant variant (Supplementary Fig. 7). Q30 outperformed wild-type AFEST: its enantioselectivity toward (*R*)-ibuprofen-*p*NP ester was decreased 2.4-fold (from $E_R = 7.2$ to $E_R = 3.0$) (Fig. 2b, Supplementary Fig. 7), and showed a 2-fold increase in the catalytic activity for (*S*)-ibuprofen-*p*NP ester.

We next used the gene encoding Q30 as a template to generate another error-prone PCR-based random mutagenesis library (Library 3). This library was comprised of 2 million variants with 2–4 mutated sites per variant. This library was screened using the DMDS cooperative mode, followed by one enrichment iteration (as above). This yielded a condensed library that comprised of ~400 variants; subsequent verification of enzyme performance in a 96-well plate assay showed that 100% of the variants had activity for (*S*)-ibuprofen (Supplementary Fig. 8c). The mutant

IE9 was identified as the top performer. It harbors an additional four mutation sites (L13H/F203E/G206E/L284F) compared to Q30, and its activity of IE9 toward (*S*)-ibuprofen-*p*NP ester is improved 2-fold (and thus cumulatively ~4-fold over wild-type AFEST). Importantly, IE9's activity toward (*R*)-ibuprofen-*p*NP ester was decreased by ~50%. As a consequence, the enantios-electivity of IE9 for (*R*)-ibuprofen-*p*NP ester was decreased 3.3-fold (from $E_R = 3.0$ to $E_S = 1.1$, Fig. 2b, Supplementary Fig. 8).

To further improve the enantioselectivity of AFEST, we expanded the screening strategy to include both increased activity toward (*S*)-ibuprofen ester and decreased activity toward (*R*)-ibuprofen ester. To accomplish this, we switched from the DMDS cooperative mode to the DMDS biased mode. The biased mode employed substrate **1** and **3** as, respectively, the selection substrate and the counter-selection substrate. Using the gene encoding IE9 as a template, we used error-prone PCR to generate a random mutagenesis library (Library 4) comprised of about 0.5 million mutant variants with an average of 2 mutation sites per variant. We screened for mutants with higher activity toward substrate **1** but with lower activity toward substrate **3**. This screening identified two positive mutants: compared to the IE9 variant, the 7F9 variant had a F90Y amino acid substitution and increased enantioselectivity for (*S*)-ibuprofen ester ($E_S = 1.1$ vs. $E_S = 1.6$); the 5G9 variant had a G88S amino acid substitution and improved enantioselectivity for (*S*)-ibuprofen ester ($E_S = 1.1$ vs. $E_S = 3.6$). These improvements in enantioselectivity were mainly due to the decreased activity toward (*R*)-ibuprofen ester, as the activity of the 7F9 and 5G9 variants toward (*S*)-ibuprofen ester remained unchanged (Fig. 2b, Supplementary Fig. 9).

Structural analysis based on the published AFEST structure revealed that AFEST residues 88 and 90 are located in Loop87–90, which is adjacent to the substrate-binding pocket (Supplementary Fig. 10); we used this information as the basis for semi-rational design to further improve the enantioselectivity based on the IE9 variant. To this end, we generated two types of libraries: single-site-saturation mutagenesis libraries (four libraries, one for each of residues 87–90) and combinational saturation mutagenesis libraries (six of these: 87/88, 87/89, 87/90, 88/89, 88/90, and 89/90). These libraries were combined into a pooled library (Library 5) comprised of about 10,000 variants that was screened using the DMDS biased mode to identify mutants with improved activity for either substrates **1** or **3**. The performance of the enzyme variants in the condensed library

**Fig. 3** Chemical structures of *rac*-**1**–**5**. Chemical names: *rac*-**1**, ibuprofen-*p*NP ester; *rac*-**2**, ketoprofen-*p*NP ester; *rac*-**3**, naproxen-*p*NP ester; *rac*-**4**, 2-phenylpropanoate-*p*NP ester; *rac*-**5**, 2-(*para*-methylphenyl)propanoate-*p*NP ester. Chemical names: *rac*-**1**, ibuprofen-*p*NP ester; *rac*-**2**, ketoprofen-*p*NP ester; *rac*-**3**, naproxen-*p*NP ester; *rac*-**4**, 2-phenylpropanoate-*p*NP ester; *rac*-**5**, 2-(*para*-methylphenyl)propanoate-*p*NP ester

were verified using 96-well plate format assays, and mutants with improved enantioselectivity toward either (S)-ibuprofen-*p*NP ester or (R)-ibuprofen-*p*NP ester were selected. Ten mutants with significantly improved enantioselectivity were obtained (Supplementary Table 3). Compared to IE9, the enantioselectivity of the top performing (S)-selective mutant variants, 6A8 (G89C/F90W on IE9) and 4D11 (G89S on IE9), was increased by, respectively, 90-fold and 72-fold (also see Supplementary Table 4 for a summary of all the key mutants throughout the evolution process and their mutation sites). Note that the enantioselectivity of the 6A8 and 4D11 variants is 700-fold and 560-fold greater than wild-type AFEST. Another interesting structural observation from this screen was that the enantioselectivity could be inverted by a single amino acid substitution: replacement of the glycine residue at the 89 position with a proline residue resulted in high enantioselectivity toward (R)-ibuprofen ester (mutant 4E12, G89P on IE9, $E_R = 74$).

Next, using several of our selected mutants, we evaluated the capability of our DMDS system to distinguish among mutant enzymes with different specificities. In the cooperative mode, mutant Q30 showed significantly improved activity compared to the wild-type enzyme toward both substrates **1** and **2**, when they were detected under the same photomultiplier gain voltages, revealing an improvement in activity (Supplementary Fig. 11). In the biased mode, mutant 5G9 showed improved activity toward substrate **1**, but decreased activity toward substrate **3**, revealing an improvement in enantioselectivity (Supplementary Fig. 12).

The (S)-selective and (R)-selective mutants were further characterized using a series of profen derivatives and their chemical structures were illustrated in Fig. 3. As shown in Table 1, for the screening substrate (*rac*-**1**), both 6A8 and 4D11 exhibited high selectivity toward the (S)-isomer with an enantiomeric excess (*ee*) value higher than 97%. The (R)-selective mutant 4E12 exhibited improved selectivity toward the (R)-isomer of ibuprofen ester, with an *ee* value as high as 95%. Moreover, we observed that both the 6A8 and 4D11 mutants showed universally increased enantioselectivity toward a series of (S)-isomers of *rac*-**2**–**5**, with the Es values from 35 to 88. The (R)-selective mutant, 4E12, also exhibited improved enantioselectivity toward different species of (R)-isomers of *rac*-**12**–**5**. These results confirm that the positive mutants obtained by using ibuprofen esters as the screening substrates show significantly enhanced enantioselectivity toward other profen derivatives as well.

**Table 1 Hydrolytic kinetic resolution of different substrates using (S)-selective and (R)-selective AFEST mutants**

| Substrate | Enzyme | Reaction time (min) | Conv. (%) | ee_p (%) | E |
|---|---|---|---|---|---|
| *rac*-**1** | WT | 10 | 23 | 70 (R) | 7.4 (R) |
| *rac*-**1** | 6A8 | 10 | 20 | >99 (S) | >100 (S) |
| *rac*-**1** | 4D11 | 6 | 18 | 97 (S) | 80 (S) |
| *rac*-**1** | 4E12 | 20 | 47 | 95 (R) | 74 (R) |
| *rac*-**2** | WT | 20 | 22 | 40 (R) | 3.0 (R) |
| *rac*-**2** | 6A8 | 30 | 24 | 79 (S) | 11 (S) |
| *rac*-**2** | 4D11 | 10 | 21 | 94 (S) | 35 (S) |
| *rac*-**2** | 4E12 | 20 | 26 | 82 (R) | 13 (R) |
| *rac*-**3** | WT | 20 | 18 | 81 (R) | 12 (R) |
| *rac*-**3** | 6A8 | 20 | 22 | 92 (S) | 31 (S) |
| *rac*-**3** | 4D11 | 10 | 34 | 93 (S) | 42 (S) |
| *rac*-**3** | 4E12 | 20 | 38 | 75 (R) | 11 (R) |
| *rac*-**4** | WT | 6 | 27 | 37 (R) | 3.0 (R) |
| *rac*-**4** | 6A8 | 60 | 29 | 68 (S) | 7.4 (S) |
| *rac*-**4** | 4D11 | 3 | 25 | 95 (S) | 88 (S) |
| *rac*-**4** | 4E12 | 20 | 22 | 68 (R) | 6.7 (R) |
| *rac*-**5** | WT | 10 | 24 | 46 (R) | 3.6 (R) |
| *rac*-**5** | 6A8 | 60 | 18 | 78 (S) | 10 (S) |
| *rac*-**5** | 4D11 | 3 | 25 | 95 (S) | 52 (S) |
| *rac*-**5** | 4E12 | 10 | 26 | 64 (R) | 6.2 (R) |

Note: The reactions were performed at 37 °C with 10 μg enzyme and 100 μM substrate in a total reaction volume of 500 μL. The reaction products were detected by HPLC equipped with a chiral column (Chiralcel OJ-H, Daicel). The mobile phases used were *n*-hexane containing 0.1% vol vol$^{-1}$ TFA and isopropanol at different ratios (90:10 for *rac*-**1**; 85:15 for *rac*-**2**, *rac*-**3**; and 95:5 for *rac*-**4**, *rac*-**5**). The total flow rate of mobile phases was 1 mL min$^{-1}$
*AFEST Archaeoglobus fulgidus esterase, HPLC high-performance liquid chromatography, TFA trifluoroacetic acid*

**Structural basis for improved enantioselectivity.** As the altered enantioselectivity of 6A8, 4D11, and 4E12 appears to result mainly from various substitutions on site 89, we used molecular docking and molecular dynamic (MD) simulations to explore the molecular mechanism underlying the improved enantioselectivity that we observed for these mutants. Molecular docking showed that although the *p*-nitrophenol group of both the (S)-ibuprofen and (R)-ibuprofen substrates were located at similar positions of the substrate-binding pocket of AFEST, the ibuprofen groups of these two compounds adopted quite different docking poses: that of (S)-ibuprofen was located at the deeper side of the substrate-binding pocket, while that of (R)-ibuprofen group was located at the entrance of the substrate-binding pocket (Fig. 4a). MD simulation revealed that the –SH group of the substitution mutant G89C resulted in severe steric hindrance against the (R)-substrate (Fig. 4b), but no obvious effect against the (S)-substrate. The G89S substitution resulted in a similar steric effect against the (R)-substrate. These results suggest the atomic basis of the improved enantioselectivity that we observed for the 6A8 and 4D11 mutants towards the (S)-substrate. Further, the substitution of G89P induced significant steric hindrance against the (S)-substrate (Fig. 4c) but no obvious effect against the (R)-substrate, which can explain the significantly increased enantioselectivity of the 4E12 mutant towards the (R)-substrate. We also found that the enantioselectivity changes of the 6A8 and 4E12 mutants were related to the methyl or phenyl ring attached to the chiral carbon atom of ibuprofen group, which is a common structure in many different types of profens.

We further explored the mechanism of 4D11 which showed higher activity and enantioselectivity than 6A8 toward *rac*-**2**, *rac*-**3**, *rac*-**4**, and *rac*-**5**. MD simulation suggested that 4D11 could form a more stable tetrahedral transition state than 6A8 during the catalysis process, rendering 4D11 more active than 6A8. For

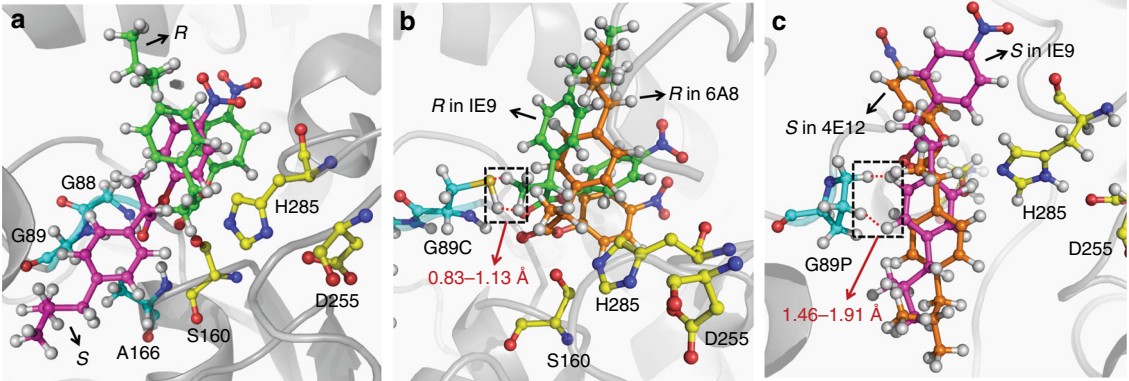

**Fig. 4** Structural basis for the altered enantioselectivity. AFEST, residues S160, D255, and H285 form a catalytic triad, while residues G88, G89, and A166 form an oxyanion hole by forming three hydrogen bonds with the carbonyl oxygen group of the (S)/(R)-substrate. **a** Docking poses of (R)/(S)-ibuprofen-pNP ester with wild-type AFEST. (R)-substrate was in a folded conformation, and the ibuprofen group was located at the substrate pocket entrance of AFEST. In contrast, (S)-substrate was in a linear conformation and the ibuprofen group was located deep inside the substrate pocket. **b** The introduction of the G89C mutation into IE9 resulted in significant steric hindrance against the chiral carbon-attaching methyl group of (R)-substrate, shifting the (R)-substrate outward. **c** The introduction of the G89P mutation into IE9 resulted in significant steric hindrance against the ibuprofen phenyl ring of (S)-substrate, forcing it to shift. The distances of the catalytic serine residue to the carbonyl group of the substrate were similar: WT-(S) 2.70 Å, WT-(R) 2.49 Å, IE9-(R) 2.89 Å, 6A8-(R) 2.62 Å, IE9-(S) 2.72 Å, and 4E12-(S) 2.87 Å

the distinct enantioselectivity toward other substrates (excepting *rac*-**1**), the F90W mutation of 6A8 made its substrate-binding site less flexible. This was not able to accommodate either larger or smaller substrates than *rac*-**1**, especially for their (S)-enantiomers. Therefore, compared to 4D11, 6A8 showed much lower activity and lower enantioselectivity toward the (S)-enantiomer of *rac*-**2**–**5**. The detailed mechanism can be found in Supplementary Fig. 13.

## Discussion

In the past decade, FADS based on microfluidics has emerged as a powerful tool for the directed evolution of enzymes[11]. This technique allows enzymatic reactions to take place within monodispersed picoliter droplets, each serving as an individual microreactor that maintains the linkage between genotype (gene encoding a target enzyme) and phenotype (typically fluorescence that signifies a reaction product). Using FADS techniques, researchers can measure enzymatic activity with accurate read-outs and then sort positive mutants in droplets, all at a throughput as high as 400–2000 droplets s per sec[11,23–31]. However, to the best of our knowledge, all of the previously reported FADS systems use only one substrate, and thus can only be applied for screening of simple enzymatic properties such as catalytic efficiency[11,23–26], enzyme expression level[23,27,28], and novel enzyme mining[29–31]. In contrast, the DMDS platform established in the present study can simultaneously evaluate enzymatic activities for two substrates. Consequently, the DMDS platform can be used in two working modes for the versatile screening of multiple enzymatic properties at the same time. The cooperative mode facilitates very highly robust and reliable screening for enzymatic activity based on two selection substrates. By exploiting the ability of the DMDS platform to monitor two completely separated reactions, the biased mode is suitable for screening complex enzymatic properties such as enantiospecificity, chemospecificity, and regiospecificity.

The successful directed evolution of AFEST variants with high enantioselectivity toward (S)-profens provides a powerful example of how our approach can be used to rapidly develop robust biocatalysts for the production of enantiomerically pure chemicals. This is highly relevant to many areas of biomedicine—for most chiral drugs, the drug administration agencies of many countries permit labeling of only a single enantiomer form[32], a reasonable rule considering that, in most cases, the curative effect

of a chiral drug results primarily from a single enantiomer while the other enantiomer can sometimes confer side effects. Considered in this light, the power of the DMDS-based directed evolution for the identification of highly enantioselective enzymes for pharmaceutical applications is readily apparent.

In recent years, a large variety of fluorogenic enzymatic assays have been developed for the high-throughput screening of enzymes of all six of the major EC catalytic categories; these developments have greatly expanded the application scope for DMDS-based directed evolution screening[33,34]. Besides the fluorogenic surrogate substrates that we used in this study, other relevant strategies include, for example, fluorescence resonance energy transfer[18], enzyme-coupled assays[35], functional group-selective reagents[36], transcriptional factors[37], riboswitches[38], and nanosensors[39], most of which can be converted with very simple modifications into dual-fluorescence assays that are amenable for use with our DMDS system (the principles for using other screening chemistries and/or biomolecules and are illustrated in Supplementary Fig. 14 and Supplementary Fig. 15, and is further discussed in Supplementary Discussion). The DMDS system is a highly versatile screening platform that can be used for the directed evolution of complex enzyme properties such as regioselectivity, chemoselectivity, and enantioselectivity.

## Methods

**The construction of *afest* libraries**. The *afest* gene was inserted into the **pUC18** vector. Libraries were created via error-prone PCR (Library 1, 3, 4), DNA shuffling (Library 2), and site-saturation mutagenesis (Library 5)[18,22,40]. Briefly, for error-prone PCR, the mutation rates were adjusted by varying the concentrations of manganese ion. The error-prone PCR system contained: DreamTaq™ (0.05 U μL⁻¹) and its buffer (Thermo), dATP (250 μM), dGTP (250 μM), dCTP (1050 μM), dTTP (1050 μM), *afest*-F (forward primer, 0.4 μM), AFEST-R (reverse primer, 0.4 μM), *afest*-**pUC18** template (0.2 ng μL⁻¹) and manganese chloride (0.2–0.8 mM). The mixture was divided into aliquots of 25 μL for error-prone PCR (95 °C for 3 min, 1 cycle; 95 °C for 15 s/55 °C for 30 s/72 °C for 1 min, 30 cycles; 72 °C for 5 min, 1 cycle). The purified PCR product was digested by *Xba*I and *Hind*III and subsequently constructed into **pUC18** vector using *T4* ligase, obtaining a random mutagenesis library. For DNA shuffling, the positive variant genes isolated from the first round of random mutagenesis library were amplified by PCR with *afest*-F and AFEST-R primers. The amplified genes were purified and mixed with the wild-type *afest* gene in a 50:1 ratio. The gene mixture was digested using benzonase (Novagen) and those DNA fragments smaller than 100 bp were collected and reassembled using the following PCR profile: 95 °C for 1 min; 45 cycles of 94 °C for 30 s, 50 °C for 30 s, 72 °C for 1 min; and 72 °C for 5 min. Analysis of the reassembly product by agarose gel electrophoresis revealed a smear encompassing the ~1000 bp target length. A portion of the assembly reaction mixture was used as template

for PCR amplification with *afest*-F and *afest*-R primers. The product were purified, digested, and ligated into **pUC18** vector, obtaining a DNA shuffling library. The site-saturation mutagenesis was achieved by whole-plasmid PCR and the primers were designed by mutating the codon of the target site with NNK (*N*-nitrosamine, 4-(methylnitrosamino)-1-(3-pyridyl)-1-butanone) (Supplementary Table 5). The whole-plasmid PCR system contained: PrimeSTAR Max DNA polymerase Premix (Takara, 25 μL), forward primer (20 μM, 1 μL), reverse primer (20 μM, 1 μL), ddH₂O (23 μL). PCR conditions: 98 °C for 3 min, 1 cycle; 98 °C for 10 s/55 °C for 10 s/72 °C for 2 min, 30 cycles; 72 °C for 5 min, 1 cycle. The template was digested by DpnI and the PCR product was purified, obtaining a site-saturation mutagenesis library. These libraries were transformed into electrocompetent *Escherichia coli* 10 G cells (Lucigen). To express the mutant enzymes, the transformed cells were grown in 10 mL 2 × YT medium containing ampicillin (0.1 mg mL$^{-1}$) for 20 h. The cells were harvested by centrifugation (5 min, 2500 *g*) and washed twice in 1 × PBS (pH 7.4). The sequences of all primers used were listed in Supplementary Table 5.

**The screening of libraries**. The cell suspension was diluted into appropriate concentrations to enable encapsulation at occupancies of 0.3, 0.1, or 0.05 cells per droplet, based on the assumption that 1 mL of this *E. coli* suspension at $A_{600\,nm} = 1$ would contain $5 \times 10^8$ cells[10]. The substrate-lysis solution contained two fluorogenic substrates (40 μM each) and a cell lysis reagent BugBuster (Novagen, 50% vol vol$^{-1}$). Our choice to use two fluorogenic substrates was made to exploit the two DMDS screening modes: substrates **1** and **2** were used for the cooperative mode, while substrates **1** and **3** were used for the biased mode (see Supplementary Fig. 5 for chemical properties, and their characterization data were shown in Supplementary Fig. 16, Supplementary Fig. 17, and Supplementary Fig. 18). Droplets were generated as described above and incubated at 37 °C for 30 min. The enzymatic reaction was terminated on ice and the droplets were injected into the detection/ sorting device. Positive droplets were sorted according to the fluorescence intensities released by the two fluorogenic substrates of a given mode. In the cooperative mode, positive events were defined as mutants with high activity toward both substrate **1** and **2** (Library 1–3). In the biased mode, positive events were defined as mutants with high activity toward substrate **1** but low activity toward substrate **3** (Library 4, 5). The sorted positive genes identified in either of the modes were then recovered by PCR, followed by restriction enzyme digestion, cloning into the pUC18 vector, and transformation into *E. coli* 10G cells for subsequent screening. Following this primary phase of DMDS screening (which can include several enrichment iterations), the positive mutants were identified by a secondary screening phase. Secondary screening was based on 96-well plate assays. To screen for catalytic activity, we used (*S*)-ibuprofen-*p*NP ester as a substrate. To screen for enantioselectivity, we used both (*S*)-ibuprofen-*p*NP ester and (*R*)-ibuprofen-*p*NP ester. These screening reactions were monitored using a plate reader (Molecular Device Inc.) at absorbance of 405 nm. The mutation sites of selected mutants were then identified by DNA sequencing. The detailed screening process—which consisted of five rounds of directed evolution—proceeded as follows:

The library for Library 1 was generated by error-prone PCR using wild-type *afest* gene as the template. This library comprised approximately 2 million variants, and the average mutation rate was 2–4 sites per variant. To achieve high throughput in the first round of DMDS screening, cells were encapsulated at high occupancy (0.3 cells per droplet). When using the DMDS cooperative mode, only 0.2% of the most active variants were retained. To achieve a 3× coverage of the total library size, we screened 18 million droplets (in <4 h). The library was thus condensed from 2 million to 12 thousand variants. In subsequent enrichment iterations, with the goal of preventing artifacts resulting from co-encapsulation at high cell occupancy, we reduced the cell occupancy to 0.05 cells per droplet. Our screen with this low occupancy retained the top 2% of the most active variants. In this manner, $7.2 \times 10^5$ mutants droplets were screened in <10 min (3× coverage of our condensed library. Finally, to validate positive mutants, 400 colonies were picked and their activity toward (*S*)-ibuprofen-*p*NP ester was monitored at 405 nm in a 96-well plate format.

The next round of directed evolution was based on DNA shuffling. There were nine mutation sites among the seven positive mutants that we selected in Library 1 (Supplementary Table 2). We shuffled these nine mutation sites, generating a DNA shuffling library (Library 2) comprised of approximately 500,000 mutants. Next, we screened Library 2 using the DMDS cooperative mode with a cell occupancy of 0.1 cells per droplet. Here, to achieve a 3× coverage, 15 million droplets were screened (in 3 h). After one round of enrichment, 200 colonies were picked, and their activity toward (*S*)-ibuprofen-*p*NP ester was monitored at 405 nm in a 96-well plate format.

Subsequently, the best mutant (Q30) identified from Library 2 was used as the template for another round of error-prone PCR-based random mutagenesis with a library comprised of approximately 2 million Q30-derived variants (Library 3); the average mutation rate was 2–4 sites per variant. Library 3 was screened using the DMDS cooperative mode, and was subjected to one enrichment iteration (same as for Library 1, above). To verify positive mutants, 400 colonies were picked and their activity toward (*S*)-ibuprofen-*p*NP ester was monitored at 405 nm in a 96-well plate format. The best mutant variant from this round was IE9.

The next round of error-prone PCR-based random mutagenesis used IE9 as the template to generate a library comprised of approximately 500,000 mutants with an average mutation rate of 1.2 sites per variant (Library 4). Library 4 was screened once with the DMDS biased mode to sort mutants with high activity toward substrate **1** but low activity toward substrate **3**, with a cell occupancy of 0.1 cells per droplet. To achieve a 3× coverage, 15 million droplets were screened (in 3 h). At

last, 200 colonies were picked from the condensed library and their enantioselectivity toward (*S*)-ibuprofen-*p*NP ester was monitored at 405 nm in a 96-well plate format.

The final screening library was a site-saturation mutagenesis library (Library 5) targeting the four AFEST residues: 87, 88, 89, and 90; both single-site and two-site combinations were generated, yielding a total of 10,000 site-saturation mutant variants. Library 5 was subjected to one round of DMDS biased mode screening to sort either (i) mutants with high activity toward substrate **1** but low activity toward substrate **3**, or (ii) mutants with high activity toward substrate **3** but low activity toward substrate **1**. The cell occupancy used in this biased mode screen was 0.05 cells per droplet. To verify the final positive mutants, 100 colonies were picked and their enantioselectivity toward (*S*)-ibuprofen-*p*NP ester or (*R*)-ibuprofen-*p*NP ester was monitored at 405 nm in a 96-well plate format.

**Computational analysis of variants**. Prior to any computational docking, we removed the sulfonyl derivative that was present in the previously reported AFEST crystal structure (PDB code 1JJI)[41] to generate what we termed the "wild-type" AFEST structure. On the basis of it, the initial structures of all the AFEST mutants were constructed and optimized by MD simulation. We next docked wild-type AFEST and mutants with the *R*-substrate and *S*-substrate which were optimized by using Gaussian 03 program[42] at B3LYP/6-31 G* level. To explore the binding mode of AFEST with the two different-conformation substrate molecules, we performed automated molecular docking by using the genetic algorithm method. During molecular docking, all the hydrogen atoms in AFEST except the polar ones were removed. The grid box around the active site of AFEST was set to the dimensions of $60 \times 60 \times 60$ Å and the grid spacing of 0.375 Å. The GA population size and the maximum number of energy evaluations were set to 150 and 250,000, respectively. All docking calculations were performed by employing AUTODOCK4.2 program[43]. Through examining the docked structures by judging the docking score, the conformation, and the amount of hydrogen bonds formed between the substrates and the active site residues, the best pose for each substrate was determined. Finally, we performed energy minimizations and MD simulations on these best poses by using AMBER9 program[44]. After adding the missing hydrogen atoms and Na$^+$ counterions to the docked structures, we solvated them in an orthorhombic box of TIP3P water molecules[45], where the minimal solute-wall distance was set as 10 Å. Then, the prepared systems were fully energy-minimized and subsequently equilibrated. The temperature in equilibration process was gradually increased from 10 to 298.15 K. The MD simulation was run for ~20 ns with the time step of 2 fs and the cutoff value for the non-bond interactions of 10 Å. The bonds containing hydrogen atom were constrained by employing Shake procedure[46,47].

**Data availability**. The NCBI accession number of the protein sequence of wild-type AFEST is WP_010879212.1. The DNA sequences of our positive mutants, 6A8, 4D11, and 4E12, have been submitted to GenBank and their accession numbers are MG925785, MG925786, and MG925787, respectively. The authors declare that the data supporting the findings of this study are available within the article and its Supplementary files, or are available from the authors upon reasonable request. The DC-FADS device was operated by a home-written code which was based on LabVIEW interface (National Instruments, USA). This code is available upon request.

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

## Acknowledgements

This work was supported by National Natural Science Foundation of China (Grant No. 21627812, 31670791, and 31470788), National Science Foundation (Grant No. CBET 1263889), and the UMich-SJTU Joint Program. F.M. was supported by China Post-doctoral Science Foundation and a Fellowship Grant to PhD Students Studying Aboard from Shanghai Jiao Tong University. M.T.C. was supported by a Graduate Program Fellowship provided by the University of Michigan Mechanical Engineering Department. R.N. was supported under the Postdoctoral Translational Scholars Program (Grant No. 2UL1TR000433). We thank Dr. Yinghua Yang as well as the Instrumental Analysis Center of SJTU for the compound characterization service.

## Author contributions

Y.F., K.K., G.-Y.Y., M.T.C. and F.M. designed the experiments. M.T.C. and R.N. designed and fabricated the microfluidic devices. L.M.L. and A.P.L. designed the droplet genera-tion assay. M.T.C. and F.M. set up the DMDS platform. F.M. and M.T.C. performed the directed evolution of *afest*. Y.Y. performed the MD simulations. G.-Y.Y., K.K., M.T.C., Y.F. and F.M. wrote the manuscript. G.Y. and K.K. directed the research.

## Additional information

**Competing interests:** The authors declare no competing interests.

