## [Peer Review File · Nature Communications]

Reviewers' comments:

Reviewer #1 (Remarks to the Author):

The manuscript describes the directed evolution of an esterase (AFEST) for improved enantioselectivity using a droplet-based microfluidics high-throughput screening technology. In comparison to previously described droplet-based microfluidics screening platforms, the key innovation of this work is the concept of dual-channel droplet sorting (DC-FADS), meaning the ability to evaluate the activity of variants against two substrates simultaneously, which allows the selection of enantioselective enzymes in the case where the substrate pair is composed of two enantiomers.

It presents a tremendous amount of work, from the development of the screening technology to its application for the directed evolution of AFEST; from the characterization of improved variants to the unveiling of the enantioselective molecular mechanisms using computational simulations.

In the current state, the manuscript is however not suitable for publication in Nature Communications. Some critical information/data are indeed lacking to fully justify the novelty and impact of the technology and understand the evolutionary route that was followed. The interest of the cooperative mode of DC-FADS is in particular not demonstrated. It is also very difficult for the reader to understand the evolutionary route that was followed (mutagenesis rounds, library sizes, screening steps, activity distribution of selected populations) and the real impact of DC-FADS technology compared to secondary microtiter plate screenings.

The following point must be addressed by the authors:

Point 1: While the power of the biased mode of the DC-FADS is obvious to screen for enantio-, chemo- or regioselectivity, the interest of the cooperative mode is less convincing. The authors claim line 71 that this mode is crucial in order to avoid the selection of false positives that are usually generated using single-substrate FADS systems (evolution of a binding site for the substrate-appended fluorophore).

Clear experimental evidence is however missing to compare both single and double-substrate FADS systems. I would suggest that the authors add an additional data set to show which improved variants emerge after the same rounds of evolution using single-substrate FADS.

Screenings of the first and second round of random mutagenesis library should be repeated using single-substrate FADS. These data are essential to point out how the evolutionary routes diverged between the two screening systems.

Point 2: The performance of the DC-FADS device was validated by sorting bar-coded droplets populations and measuring the fluorescence signals of sorted and unsorted populations (Figure S4). In both presented cases, the rate of false negatives (unsorted positive droplets) was found to be 0.2% and the rate of false positives (sorted negative droplets) was found to be 3-4%. This rate of false positives is rather high when selecting for rare events (2% in the first round, 0.2% in the second round). It is worth noting that additional false positives are likely to be selected when encapsulated cells are sorted because of co-encapsulation events, as described in Baret et al; (Lab Chip, 2009, 9, 1850-1858).

- can the authors comment on this high false positive rate? Were the DC-FADS performance experiments performed at 1400 droplet/s? Why not decreasing the sorting frequency to reach a higher degree of robustness?

- How does this 3-4% error rate compare to the rate of false positives generated by single-substrate FADS, which are supposed to be avoided by the "cooperative mode"?

- The authors should mention the first historical paper introducing FADS technology (Baret et al; Lab Chip, 2009, 9, 1850-1858).

Point 3: Experimental details are lacking regarding the screening process. What is the off-chip incubation time of the droplets before sorting?

Point 4: Related to point 3, do the authors have experimental data about the transport properties of the fluorescent dye 1, 7-hydroxycoumarin-3-carboxylic acid (HCCA)? To the best of my knowledge, coumarin derivatives tend to be easily transported in water in fluorinated oil emulsions. Introduction of highly hydrophobic moieties, such as sulfonate groups, allows increasing retention times and limiting leakage of these molecules from droplets to droplets (Woronoff et al., Anal. Chem, 2011, 83, 2852-2857). The log D of HCCA is -2.41 (Ma et al., Anal. Chem, 2016, 88, 8587-8595), which indicates that it might be transported through the oil phase. Can the authors provide data about the half-time of retention of HCCA in their water-in-fluorinated oil system? How does this retention time compare to the emulsion incubation time asked in point 3?

Point 5: The description of the iterative rounds of evolution and screening is not clear for the reader. The first 2 million-mutants library was subjected to two screening rounds followed by a secondary microtiter plate screening. How many variants were screened in microtiter plate? Seven positive mutants were identified and used to create a DNA shuffling library. What is the size of this library? What is the screening process that led to mutant Q30 (droplet, well)?

Point 6: In general, how many variants were screened in each secondary microtiter plate screening (first, second and third round of mutagenesis)? During these secondary screenings, what is the proportion of active variants? These points have to be clarified in order to understand the real role of the DC-FADS first screenings. For each secondary microtiter plate screening, the authors should provide the number of assayed clones and the activity distribution (activity of each assayed clones against wild type activity). This will give important information about the proportion of active/inactive variants selected after each DC-FADS screenings.

Point 7: The first round of screening (Figure S6) is performed in the cooperative mode to select variants that show activity toward both substrate 1 and 2. The activity distribution of the second round library (Figure S7a) shows however that most of the variant are more active on the Coumarin-(S)-ibuprofen (the 2D plot distribution not anymore align on the 1-1 diagonal). How do the authors explain this selectivity? Shouldn't it be the kind of activity profile that is supposed to be avoided using the dual-channel cooperative mode?

Point 8: In Figure S8, in line 149 is mentioned that the most active mutants toward both substrates were sorted, while the screening is supposed to be in the biased mode.

Reviewer #2 (Remarks to the Author):

When composing your report, the following questions might assist you in writing an incisive, well-justified review. What are the major claims of the paper? Are they novel and will they be of interest to others in the community and the wider field? If the conclusions are not original, it would be helpful if you could provide relevant references. Is the work convincing, and if not, what further evidence would be required to strengthen the conclusions? On a more subjective note, do you feel that the paper will influence thinking in the field? Please feel free to raise any further questions and concerns about the paper.

We would also be grateful if you could comment on the appropriateness and validity of any statistical analysis, as well the ability of a researcher to reproduce the work, given the level of detail provided.

Review Manuscript NCOMMS-17-24439

This manuscript describes a new way to screen mutant libraries of enzymes very fast for improved enantioselectivity. The dual-channel microfluidic droplet screening (DMDS) that is described in this paper is to my knowledge a novel concept and interesting for other scientists in biocatalysis, but also in other sciences, like pharmaceuticals.

Until now, fast screens using FACS and 96-well plates usually rely on a double screening effort (one enantiomer at the time), while the concept in this manuscript is using racemic mixtures which is closer to the application of these enzymes. The authors continue on the double fluorescence substrates to reduce false positives (ref. 11). The new idea of screening in two modes is imperative when screening for improved enantioselectivity. Therefore, it could produce the better variants much faster than with state-of-the-art screening methods. The two modes are also important in order to keep improving the enzymes. Just screening for the same thing over and over again, doesn't result in better enzymes as is also shown in this paper (not much difference in activity in M2 although E does differ). The combination of different screening methods and different mutant library generating methods results in the final enzymes with medium to excellent enantioselectivity, which is pretty good considering the effort.

Although I do like the manuscript and the concise way it is written, I do have a few questions and remarks to improve the paper.

It is not clear of all random mutagenesis libraries how they were constructed (epPCR or also other techniques used?). Also the mutation rate of library 2 is not mentioned. Please clarify.

In table 1 there are other structures drawn, but both in the table and in the text there are no names mentioned of these structures. For me as a non-chemist, this would be helpful.

About table 1 and the text describing it: I see that except for ibuprofen (the first entry), mutant 4D11 is always more active and more enantioselective than mutant 6A8. This is probably the reason why the authors added it to the paper. Unfortunately, nothing is really said about it in the text, while it would be really interesting to understand why that mutant is better on the other substrates. There is also no picture of this one in figure 3. It would be nice to have a few more words on this mutant and the fact that the best mutant on the original screening substrate is not always the best on similar substrates.

In figure 3 it would also be helpful to add the distances of the catalytic Ser to the bond that should be attacked, or if it is the same everywhere, please mention this number in the text or legend of the figure.

Although the results are impressive, I am also wondering how generic this screen is. It will probably work for other enzymes that have a hydrolytic activity. And since this is an important class of enzymes, it will definitely be used by others. But what about other enzymes? Maybe it is outside the scope of this paper and it is just a question for other researchers to continue on this research, but I am wondering how to screen non-hydrolytic enzymes (like lyases or ligases).

About the references: I think the first 6 references (on improving enantioselectivity by directed evolution) are a bit old. I understand you want to show some original references, but it would be better to add one or more recent ones as well, like: C.G. Acevedo-Rocha et al. 2010 (doi: 10.1016/j.jbiotec.2014.04.009), Otten et al. 2010 (doi: 10.1016/j.tibtech.2009.10.001), Reetz et al. 2010 (doi: 10.1351/PAC-CON-09-09-16), Denard et al. 2015 (doi: 10.1016/j.cbpa.2014.12.036), Xiao et al. 2015 (doi: 10.1021/ie503060a), Autour and Ryckelynck 2017 (doi: 10.3390/mi8040128), Reetz 2017 (doi: 10.1007/978-3-319-50413-1_3), Longwell et al. 2017 (doi: 10.1016/j.copbio.2017.05.012).

Figure S5 is a bit strange. The synthesis of substrate 1 is in panel B, while the synthesis of substrates 2 and 3 are in panel A. Maybe this can be turned around for clarification?

In figures S6, S7 and S8 I had some problems in understanding the differences between the red and green bars. Maybe it can be added to the legend that red is activity (left axis) and green is enantioselectivity (right axis).

Minor things

There are some typing mistakes in the following lines, including the omission of many articles, which should be changed: 11, 16, 23 87, 167, 179, 188, 205, 206, 207, 208, 211, 212, 213.

Typing mistakes in the SuppInfo are on lines: 62, 92, 143.

In line 185 it says: ..higher than 99%. Mutant 6A8 is indeed >99%, but mutant 4D11 is 97%, so this should be: ..higher than 97%.

In line 162 and 163, the authors mention mutant 7A5, but I think this should be 7F9 (if this is correct, than fig. 2 is wrong). Also check Table S1 and S2 on this issue.

Reference 3 and 6 are (almost) duplicates. Ref. 6 is correct, ref. 3 has the wrong year. Maybe the authors wanted to put this reference on position 3: Reetz et al. 2010 (doi: 10.1351/PAC-CON-09-09-16)?

In conclusion: I recommend publication with minor revisions.

Reviewer #3 (Remarks to the Author):

This paper describes a new screening protocol for directed evolution. This is a very important area because the most time consuming and prohibitive part of improving enzymes is the screening of the millions of mutants that are generated.

Overall I find the manuscript describe some very interesting science; however, it isn't clear to me how this revolutionizes the directed evolution process. This process seems very enzyme-substrate specific. How would this process work for a ketoreductase reaction, etc? In particular, I would expect most enzymes would not take a substrate that has a large dye attached to it due to active site size vs substrate size. As a result, the way it is written, I do not feel like this manuscript rises to the proper level to be accepted in this journal.

Comments:

This article seems to be written out of order. I was massively confused for the first three pages. It seems like the narrative should start out at line 106 (this is where I started to get a better idea of their work and what was done)... and it seemed to be written more clearly (authors changed?) Similarly, they use the terms cooperative and biased mode and then don't define those terms until line 80. I suggest the authors step back and rethink how they want to explain this work. Perhaps, when better described, I would then better appreciate the impact that they feel this work will have on the improvement of the field of biocatalysis and enzyme evolution.

Side note: I collaborated with a computational chemist who did similar MD simulations (using autodock, etc) and many reviewers hated this work and said that this type of simulations weren't accurate enough to be published? It is outside of my field so I will leave it up to the editor to find a reviewer to comment on that portion.

Response to Reviewers' comments:

We greatly appreciate the helpful comments from all the reviewers on our manuscript entitled "**Efficient molecular evolution to generate enantioselective enzymes using a dual-channel microfluidic droplet screening platform**". Our point-by-point responses to the reviewers are presented below. Guided by these reviewer comments, we have made corrections and improvements to our original manuscript. **The major revisions we have made to this manuscript include:**

- 1) We have comprehensively redrafted the *abstract, introduction, results, and discussion* sections to make the manuscript more readable and logical.
- 2) We have emphasized the efficiency of the *cooperative* mode of our DMDS system in the revised manuscript.
- 3) We have enhanced the mechanistic information about the atomic basis of the observed increase in enantioselectivity of AFEST in both the manuscript and the Supplementary Information.
- 4) We have discussed the versatility of the DMDS system for screening complicated properties of various enzymes in *discussion* section, as well as in the Supplementary Information.
- 5) To make the directed evolution experiment more understandable, we have added the details of the library construction and screening process to *methods* section and now also number the libraries as we introduce them in the main argument.
- 6) Various modifications have been made to meet the requirements of the editorial policies of Nature Communications.

Our responses to the reviewers' comments

Reviewer #1:

Point 1: While the power of the biased mode of the DC-FADS is obvious to screen for enantio-, chemo- or regioselectivity, the interest of the cooperative mode is less convincing. The authors claim line 71 that this mode is crucial in order to avoid the selection of false positives that are usually generated using single-substrate FADS systems (evolution of a binding site for the substrate-appended fluorophore).

Clear experimental evidence is however missing to compare both single and double-substrate FADS systems. I would suggest that the authors add an additional data set to show which improved variants emerge after the same rounds of evolution using single-substrate FADS. **Screenings of the first and second round of random mutagenesis library should be repeated using**

single-substrate FADS. These data are essential to point out how the evolutionary routes diverged between the two screening systems.

Response: We thank the reviewer for this helpful discussion point. The so-called first law of directed evolution states: “*you get what you screen for*”, that is, the condition used in a given screen has an enormous influence on the screening results (*Schmidt-Dannert et al. Trends Biotechnol. 1999, 17:135–136*). Numerous studies have also shown that false positives can arise when indirect measurements are used in the screens rather than monitoring the targeted activity directly (*Wahler et al., Curr Opin Chem Biol. 2001, 5(2):152-158. Tholey et al., Adv Biochem Eng Biotechnol. 2002, 74:1-19*). For example, when a single fluorogenic surrogate substrate was used for the screening of a glycosyltransferase, several false positive mutants were identified because of the evolution of a binding site for the substrate-appended fluorophore (*Aharoni et al., Nature Methods, 2006, 3(8):609-614*). However, such screens can be improved by using two fluorophores simultaneously (*Yang et al., J Am Chem Soc. 2010, 32(30):10570-10577*); this is because it is extremely unlikely for a mutant to evolve a binding site for two chemically distinct fluorophores simultaneously. A two-fluorophore strategy thus minimizes the possibility of isolating false positives from huge mutation libraries.

Similarly, in the *cooperative* mode of our DMDS platform, the activities of mutants of the target protein were measured using two separate fluorogenic substrates, simultaneously, to minimize the generation of false positive mutants. As the DMDS system represents a general screening platform for any fluorescent enzymatic reaction, we believe its *cooperative* mode has the potential to be used for the high-fidelity screening of many enzymes, as long as proper fluorescent substrates are available (for more discussion regarding this issue please see the response to Point 5 of the second reviewer).

However, it should be noted that the emergence of a false positive does not necessarily occur in each round of evolution. It is a case-specific phenomenon that depends on which enzyme and what kind of property you are screening for. So, even if we were to attempt to somehow repeat the entire screening process using single-substrate FADS, it is not clear that we could expect any false positives to occur. However, we have every confidence that as we continue to test the DMDS system with a variety of enzymes and different libraries, the advantages of DMDS system over single-substrate FADS will be obvious; the types of assays one can do with a single-channel vs. a dual-channel system are almost completely different/orthogonal. As the reviewer suggested, biologically, it would indeed be very interesting to study the evolutionary route of this esterase under a single-substrate FADS and under the *cooperative* mode of DMDS screening, yet this basic line of inquiry is well beyond the scope of the technological development that is the core of the study presented in this manuscript. Nevertheless, we have added exciting mechanistic information about the atomic basis of the observed increase in enantioselectivity of AFEST to

bolster the biochemistry content in the revised manuscript (page 12, line 310-318).

Point 2: The performance of the DC-FADS device was validated by sorting bar-coded droplets populations and measuring the fluorescence signals of sorted and unsorted populations (Figure S4). In both presented cases, the rate of false negatives (unsorted positive droplets) was found to be 0.2% and the rate of false positives (sorted negative droplets) was found to be 3-4%. This rate of false positives is rather high when selecting for rare events (2% in the first round, 0.2% in the second round). It is worth noting that additional false positives are likely to be selected when encapsulated cells are sorted because of co-encapsulation events, as described in Baret et al; (Lab Chip, 2009, 9, 1850-1858).

- **can the authors comment on this high false positive rate? Were the DC-FADS performance experiments performed at 1400 droplet/s? Why not decreasing the sorting frequency to reach a higher degree of robustness?**
- **How does this 3-4% error rate compare to the rate of false positives generated by single-substrate FADS, which are supposed to be avoided by the "cooperative mode"?**
- **The authors should mentioned the first historical paper introducing FADS technology (Baret et al; Lab Chip, 2009, 9, 1850-1858).**

Response:

Thank you for bringing these issues to our attention. First, let us note that we have significantly clarified the definitions for the "false positives" that we mention in the revised manuscript. In this case in particular, the false positive was caused by the mistaken collection of non-fluorescent droplets. This is distinct from what we discussed in Point 1, wherein the droplet gave a signal from an enzyme variant with high activity toward the screening substrate but not the real target substrate. Thus, this particular false positive was caused by the inaccuracy of the sorting device itself.

Secondly, it is important to note that the false positive rate here was calculated in a different way from that of Baret *et al.*'s paper. What Baret *et al.* reported was that one non-fluorescent droplet would be collected from among 10,000 **detected** droplets using their FADS system (Baret *et al.*, Lab Chip, 2009, 9, 1850-1858). However, what we reported is that 3-4 droplets that have no fluorescence would be present in 100 **collected** droplets. Given the positive ratio before sorting in our study (10% and 1%), the false positive rate in our

DMDS system (in total droplets before sorting) is low (~ 3 in 10,000 droplets), which is comparable to the system reported by Baret *et al.* This has been clarified in the revised manuscript; indeed, we now use the calculations that Baret *et al.* used in our revised text. We are also thankful for the reviewer's suggestion about the first historical paper introducing FADS technology (Baret *et al.*, *Lab Chip*, 2009, 9, 1850-1858), we have added it to the revised manuscript (page 4, line 110-113).

Point 3: Experimental details are lacking regarding the screening process. **What is the off-chip incubation time of the droplets before sorting?**

Response: We thank the reviewer for the helpful comments. The off-chip incubations were for 30 min at 37 °C. We have comprehensively rewritten the *Methods* section (page 14-16, line 367-438) detailing the screening and we have added important clarifying details to the *Results* section as well. We have carefully selected the terms that we use to delineate the separate types of libraries (and also numbered the libraries) and have been very methodical in our step-by-step presentation of how these screens proceeded.

Point 4: Related to point 3, do the authors have experimental data about the transport properties of the fluorescent dye 1, 7-hydroxycoumarin-3-carboxylic acid (HCCA)? To the best of my knowledge, coumarin derivatives tend to be easily transported in water in fluorinated oil emulsions. Introduction of highly **hydrophobic** (should be hydrophilic) moieties, such as sulfonate groups, allows increasing retention times and limiting leakage of these molecules from droplets to droplets (Woronoff *et al.*, *Anal. Chem*, 2011, 83, 2852-2857). The log D of HCCA is -2.41 (Ma *et al.*, *Anal. Chem*, 2016, 88, 8587-8595), which indicates that it might be transported through the oil phase. **Can the authors provide data about the half-time of retention of HCCA in their water-in-fluorinated oil system? How does this retention time compare to the emulsion incubation time asked in point 3?**

Response: In general, the lower log D value, the more hydrophilic a compound is, and the better droplet retention it will have (Woronoff *et al.*, *Anal. Chem*, 2011, 83, 2852-2857). The HCCA fluorophore we used in our work is a coumarin derivative with relatively strong hydrophilicity. It has been demonstrated that the critical log D value for satisfying retention is -2.13 in mineral oil based emulsion system (Ma *et al.*, *Anal. Chem*, 2016, 88, 8587-8595). HCCA, with its

log D value of -2.41, has higher hydrophilicity than would be required to be well retained in mineral oil droplets. Since the fluorinated oil system that we used has better retention than a mineral oil system (*Holtze et al., Lab Chip 2008, 8, 1632–1639.*), the hydrophilicity of HCCA is more than adequate to be stable within the reaction and screening time frame of our experiments. Indeed, our experiment also revealed that more than 95% of fluorescence can be retained within 30-min of off-chip incubation (data not shown). To clarify this point, we have added discussion of the suitability of HCCA in droplet-based assays in the “*Synthesis of fluorogenic substrates derived from ibuprofen*” part of the revised Supplementary Information (page S8, line 81-86).

Point 5: The description of the iterative rounds of evolution and screening is not clear for the reader. The first 2 million-mutants library was subjected to two screening rounds followed by a secondary microtiter plate screening. **How many variants were screened in microtiter plate? Seven positive mutants were identified and used to create a DNA shuffling library. What is the size of this library? What is the screening process that led to mutant Q30 (droplet, well)?**

Response: We apologize for the obscure description on the screening process; we now appreciate the lack of clarity in the initial submission and (as mentioned below in response to Point 6) we have comprehensively rewritten this part of the methods and made extensive changes and clarifications in the results text as well. In specific response to this comment: for the first error-prone PCR based library (Library 1) of ~2 million mutant variants of AFEST, 400 variants were screened in 96-well microtiter plates, and seven positive mutants were selected for further study (Table S2 in the revised Supplementary Information). Subsequently, a DNA shuffling library was created using the 9 mutations present among the seven selected positive mutants, with a library size of 500,000 mutants. The DNA shuffling library was subjected to one round of DMDS screening (in the *cooperative* mode) and then 200 putatively positive mutants were randomly selected for a secondary screening in microtiter plates: the Q30 variant was thusly identified as the top performing mutant (also see Table R1 in the response to Point 6 for the summary of the entire screening process). More details about the whole screening process have been added in the *Materials* part of the manuscript (page 14-16, line 367-438).

Point 6: In general, how many variants were screened in each secondary microtiter plate screening (first, second and third round of mutagenesis)? During

these secondary screenings, what is the proportion of active variants? These points have to be clarified in order to understand the real role of the DC-FADS first screenings. **For each secondary microtiter plate screening, the authors should provide the number of assayed clones and the activity distribution (activity of each assayed clones against wild type activity). This will give important information about the proportion of active/inactive variants selected after each DC-FADS screenings.**

Response: We are thankful for the review's helpful discussions. We have included this information in the revised text. Additionally, for convenience, the number of variants screened in each secondary microtiter plate is summarized below in Table R1. And the activity distribution of the assayed clones in these secondary screenings is illustrated in Figure R1. These data show that most of the clones are active (defined here as >50% of the wild-type activity as a threshold), demonstrating the robustness of the *cooperative* mode to enrich positive mutants.

Table R1. Summary of the whole screening process.

Library number	Library type	Template	Library size	DMDS Screening mode	Number of variants for secondary screening	Proportion of active variants ^b
1	Ep-PCR ^a	WT AFEST	2,000,000	cooperative	400	99%
2	DNA shuffling	7 mutants from Library 1	500,000	cooperative	200	100%
3	Ep-PCR	Q30	2,000,000	cooperative	400	100%
4	Ep-PCR	IE9	500,000	biased	200	n.d. ^c
5	site-saturation mutagenesis	IE9	10,000	biased	100	n.d. ^c

^a Ep-PCR: Error-prone PCR

^b The proportion of active variants was identified as the ratio of mutants which showed activities >50% of the wild-type AFEST.

^c n.d. Not determined. This is because Library 4 and Library 5 were screened based on changes in enantioselectivity; so some of the variants that had higher enantioselectivity but overall lower enzyme reaction activities were also included in the secondary screening.

Figure R1. The proportion of active variants after DMDS screening. (a) Mutants enriched from the Library 1. (b) Mutants enriched from Library 2. (c) Mutants enriched from Library 3. The active variants were defined as the mutants with activities >50% of the wild-type AFEST.

Point 7: The first round of screening (Figure S6) is performed in the cooperative mode to select variants that show activity toward both substrate 1 and 2. The activity distribution of the second round library (Figure S7a) shows however that most of the variant are more active on the Coumarin-(S)-ibuprofen (the 2D plot distribution not anymore align on the 1-1 diagonal). **How do the authors explain this selectivity? Shouldn't it be the kind of activity profile that is supposed to be avoided using the dual-channel cooperative mode?**

Response: Thanks, this is an excellent question. In the DMDS system, the fluorescence of reaction products is detected using a photomultiplier tube (PMT), which converts a fluorescence signal into an electrical signal; its sensitivity can be adjusted by altering detection voltages. To ensure that the detection window used for each of the successive mutant libraries was suitable, we adjusted the detection voltages manually for each round of screening. Consequently, the activity distributions in different rounds of screening are not directly comparable because they use different detection voltages. Therefore, the difference between Figure S6 and S7 in activity distribution toward substrate **1** and **2** was actually caused by different PMT detection voltages as applied in the two separate tests.

When the wild-type AFEST and its mutants were measured under the same PMT voltages, the capability of DMDS system in distinguishing different enzymes can be clearly demonstrated. As shown in Figure R2, mutant Q30 shows significantly improved fluorescence toward both substrate **1** and **2**, revealing an improvement in activity in the *cooperative* mode. In the *biased* mode, mutant 5G9 shows improved fluorescence toward substrate **1**, but decreased fluorescence toward substrate **3**, revealing an improvement in enantioselectivity

(Figure R3). To make this more clear, we further explained the details of the PMT detection voltages in the revised Supplementary Information (the legend of Figure S7). We also incorporate the comparison of WT AFEST and its mutants under the same PMT detection voltages to demonstrate the capability of DMDS system in distinguishing different enzymes (Revised *Results* section in the manuscript, page 10, line 246-252, as well as Figure S11, S12 in Supplementary Information).

Figure R2. Evaluation of enzymatic activity of wild-type AFEST (a) and Q30 (b) in **cooperative** mode. Q30 showed significantly improved activities toward both coumarin-(S)-ibuprofen (substrate **1**) and fluorescein-(S)-ibuprofen (substrate **2**). WT AFEST and Q30 were detected under the same PMT voltages.

Figure R3. Evaluation of enzymatic enantioselectivity of wild-type AFEST (a) and 5G9 (b) in **biased** mode. Compared with wild-type AFEST, 5G9 showed significantly increased activities toward coumarin-(S)-ibuprofen (substrate **1**) but decreased activity toward fluorescein-(R)-ibuprofen (substrate **3**), which was in accordance with its improved enantioselectivity toward (S)-ibuprofen substrates. WT AFEST and 5G9 were detected under the same PMT voltages.

Point 8: In Figure S8, in line 149 is mentioned that the most active mutants toward both substrates were sorted, while the screening is supposed to be in the biased mode.

Response: We apologize for this mistake and are grateful for the careful review. Indeed, the screening in Figure S8 was in the *biased* mode. This has been corrected in the revised manuscript.

Reviewer #2:

Point 1: It is not clear of all random mutagenesis libraries how they were constructed (epPCR or also other techniques used?). Also the mutation rate of library 2 is not mentioned. Please clarify.

Response: We appreciate the reviewer's helpful suggestion here, and have clarified this in several places throughout the revised manuscript. To specifically answer the present question: the mutation rate of "library 2" was between 2-4 sites per gene variant. The random mutagenesis libraries were constructed by error-prone PCR, but we also generated a DNA shuffling library and several site-saturation mutation libraries. We have taken considerable care in adding this obviously important information to the results and methods sections throughout the revised manuscript.

Point 2: In table 1 there are other structures drawn, but both in the table and in the text there are no names mentioned of these structures. For me as a non-chemist, this would be helpful.

Response: We appreciate the reviewer's comment and have addressed it. We have added the chemical names of these compounds in the footnotes of revised Table 1.

Point 3: About table 1 and the text describing it: I see that except for ibuprofen (the first entry), mutant 4D11 is always more active and more enantioselective than mutant 6A8. This is probably the reason why the authors added it to the paper. Unfortunately, nothing is really said about it in the text, while it would be really interesting to understand why that mutant is better on the other

substrates. There is also no picture of this one in figure 3. It would be nice to have a few more words on this mutant and the fact that the best mutant on the original screening substrate is not always the best on similar substrates.

Response: We are thankful for the reviewer's constructive suggestions. We have now investigated the mechanism underlying the comparatively stronger activity and enantioselectivity of the 4D11 mutant for other substrates relative to the 6A8 mutant.

First, molecular docking and molecular dynamic simulation showed that both the G89C mutation of the 6A8 mutant protein and the G89S mutation of the 4D11 mutant protein can form hydrogen bonds with the carbonyl oxygen of the substrates. The hydrogen bond can stabilize the tetrahedral transition state during the catalysis process. However, the hydrogen bond O-H \cdots O formed by the G89S mutation of 4D11 is stronger than the S-H \cdots O formed by the G89C mutation of 6A8, suggesting that 4D11 can form more stable tetrahedral transition state than 6A8. This may explain why 4D11 has higher activity than 6A8 toward all substrates tested.

Second, as shown in Figure R4a, the G89S/C mutations of 6A8 and 4D11 introduce steric hindrance against (*R*)-ibuprofen-*p*-nitrophenol ester but confer no obvious effect on (*S*)-ibuprofen-*p*-nitrophenol ester, thereby making these mutants (*S*)-enantioselective. The additional F90W mutation on 6A8 enhances the pi-pi interaction with Y222. On one hand, this effect stabilizes the oxyanion hole structure (G88, G89), which is beneficial for the catalysis process of 6A8. On the other hand, this effect decreases the flexibility of the substrate-binding pocket, making 6A8 less active for substrates that are either smaller or larger than (*S*)-ibuprofen-*p*-nitrophenol ester. For example, *rac-2* and *rac-3* have larger sized profen moieties, and the substrate-binding pocket of 6A8 (with less flexibility) may cause steric hindrance against their (*S*)-enantiomers. This appears to decrease the enantioselectivity of 6A8 toward *rac-2* and *rac-3*. Figure R4b illustrates that the profen moiety of (*S*)-ibuprofen-*p*-nitrophenol ester is surrounded by many hydrophobic residues, forming hydrophobic interactions that are beneficial for enzyme-substrate recognition. Since the profen moieties of *rac-4* and *rac-5* are smaller than that of *rac-1*, the hydrophobic interactions between the substrate-binding pocket and their (*S*)-enantiomers should be significantly diminished due to the low flexibility of 6A8. This explains how the mutant 6A8 has lower enantioselectivity than 4D11 toward *rac-4* and *rac-5*.

Figure R4. Comparison of mutant 6A8 and 4D11, which exhibit distinct enantioselectivity toward various substrates. **a** Docking poses of (*R*)/(*S*)-ibuprofen-*p*-nitrophenol ester with 6A8 and 4D11. Mutation G89C of 6A8 introduced significant steric hindrance against (*R*)-ibuprofen-*p*-nitrophenol ester (the shortest distance between the -SH group of the mutation and the substrate was 0.83 Å), but no obvious steric hindrance against (*S*)-ibuprofen-*p*-nitrophenol ester (the shortest distance was 2.2 Å). Similar steric hindrance was also introduced by the G89S mutation of 4D11. Further, the F90W mutation of 6A8 enhanced the pi-pi interaction with Y222, which stabilized the oxyanion hole structure (G88, G89, A166) yet meanwhile decreased the flexibility of the substrate pocket. **b** The profen moiety of the (*R*)-ibuprofen-*p*-nitrophenol ester is surrounded by hydrophobic residues (in orange), forming hydrophobic interactions.

We have added the content regarding the atomic basis of the performance for the 4D11 mutant variant to the revised manuscript: the mechanism of 4D11 with better properties than 6A8 is briefly discussed in the *Results* part of manuscript (page 12, line 310-318), and the detailed information is now shown in the revised Supplementary Information (Figure S13).

Point 4: In figure 3 it would also be helpful to add the distances of the catalytic Ser to the bond that should be attacked, or if it is the same everywhere, please mention this number in the text or legend of the figure.

Response: We appreciate the reviewer's helpful suggestion here. The distances between the catalytic Serine (Ser160) to the carbonyl group of the substrates were similar: WT-(*S*) 2.70 Å, WT-(*R*) 2.49 Å, IE9-(*R*) 2.89 Å, 6A8-(*R*) 2.62 Å,

IE9-(S) 2.72 Å, 4E12-(S) 2.87 Å. We have added the distances between the catalytic Serine to the carbonyl group of the substrates in the legend of Figure 3.

Point 5: Although the results are impressive, I am also wondering how generic this screen is. It will probably work for other enzymes that have a hydrolytic activity. And since this is an important class of enzymes, it will definitely be used by others. But what about other enzymes? Maybe it is outside the scope of this paper and it is just a question for other researchers to continue on this research, but I am wondering how to screen non-hydrolytic enzymes (like lyases or ligases).

Response: This is a very important topic to discuss. The DMDS system provides a platform for the detection and sorting of two fluorescent signals simultaneously in micro-droplets at a speed of more than one million reads per hour. However, it is equally important to develop fluorogenic assays for the enzymatic reactions of interest. In recent years, a large variety of fluorogenic enzymatic assays have been developed for the high throughput screening of enzymes of all six of the major EC catalytic categories (Reymond *et al.*, *Chem Commun (Camb)*. 2009, (1):34-46. Goddard *et al.*, *Trends Biotechnol.* 2004, 22(7):363-70.). Besides the fluorogenic surrogate substrates that we used in this study, other strategies include fluorescence resonance energy transfer (FRET) (Yang *et al.*, *Angew Chem Int Ed Engl.* 2015, 54(18):5389-5393.), enzyme-coupled assays (Kumagai *et al.* *Anal Biochem.* 2014, 447:146-155.), functional group selective reagents (Gotor *et al.*, *Anal Chem.* 2017, 89(16):8437-8444), transcriptional factors (Mahr *et al.*, *Appl Microbiol Biotechnol.* 2016, 100(1):79-90.), riboswitches (Su *et al.*, *J. Am. Chem. Soc.*, 2016, 138(22), 7040–7047) and nano-sensors (Zhang *et al.*, *Biosens Bioelectron.* 2013, 44:6-9.). Most of these assays can (in theory) be used in the DMDS system with very simple modifications, thus opening up the possibility for ultrahigh-throughput screening for a great many enzymatic reactions indeed.

To answer the reviewer's question, current technology allows the screening of both **ligases** and **lyases** using DMDS. For example, a FRET-based assay has been developed for DNA ligases (Shapiro *et al.*, *J Biomol Screen.* 2011, 16(5):486-493) and for Ubiquitin ligases (Sun *et al.*, *Methods Enzymol.* 2005, 399:654-663.). As illustrated in Figure R5a (below), it is possible to link two discrete substrate fragments to separate moieties (e.g., a FRET donor and FRET quencher). Different FRET signals then can be used to screen for the substrate specificities for ligases.

Fluorescence assays have also been developed for **lyases**. Taking aldolases as an example, the fluorogenic aldol substrates have been designed such that it can be catalyzed by the β -elimination activity of aldolases to give 3-oxopropyl

umbelliferyl ether, which is subsequently converted into umbelliferone via a spontaneous β -elimination reaction (Nathalie et al., *Tetrahedron Lett* 39 (1998) 9415-9418. Pérez et al., *Chemistry*. 2000 Nov 17;6(22):4154-62). As illustrated in Figure R5b, two fluorogenic substrates can be designed by using two aldol enantiomers that are conjugated to different fluorophores. After enzymatic reaction, substrate selectivity can be evaluated by measuring the fluorescence ratios between two fluorophores (similar with the case of hydrolases).

A similar strategy has been used for the screening of C-C bond cleavage **transferases** such as transaldolases (González et al., *Chemistry*. 2003, 9(4):893-899.) and transketolase (Sevestre et al., *Tetrahedron Lett*, 2003, 44 (4):827-830).

Note that the DMDS system can be also used for the screening of ketoreductases, an important class of **oxidoreductase** enzyme. Truppo et al. (2008) developed a fluorescent assay for the enantioselectivity of ketoreductases in microtiter plates (Truppo et al., *Angew Chem Int Ed Engl*. 2008, 47(14):2639-2641.). This strategy can be directly used in the screening for the enantioselectivity of ketoreductases in our DMDS system. As illustrated in Figure R5c, the (*R*)-alcohol product could be detected specifically by coupling an (*R*)-selective alcohol oxidase, generating resorufin with red fluorescence. Meanwhile, the consumption of NADPH which represents the total production of (*R*)- and (*S*)-alcohol products is detected by measuring the decrease of NADPH fluorescence (Blacker et al., *Nat Commun*. 2014, 5:3936-3944. Patterson et al., *Proc Natl Acad Sci U S A*. 2000, 97(10):5203-5207.). The enantioselectivity of the ketoreductase can be calculated as the ratio between coumarin fluorescence and decreased NADPH fluorescence.

Furthermore, with the recent fast development of biosensors (such as transcriptional factors or riboswitches), it is possible to convert the concentration of any targeted compound into the overexpression of downstream reporter gene (e.g. fluorescence proteins) (Eggeling et al., *Curr Opin Biotechnol*. 2015, 35:30-36.). As illustrated in Figure R6, it is possible to design artificial biosensors for any enzymatic products of interest to activate various reporter genes, which will greatly promote the application of the DMDS system. (Michener et al., *Metab Eng.*, 14 (2012) 306-316. Wang et al., *Biotechnol Bioeng*. 2016 Jan;113(1):206-15. Looger et al., *Nature*. 2003, 423(6936):185-190.).

Figure R5. Schematics of dual-fluorescence assays for screening selectivity of enzymes using the DMDS system. **a**, A strategy for screening selectivity of ligases. **b**, A strategy for screening enantioselectivity of aldolases. The dye

moieties may be coumarin or resorufin derivatives (*List et al., Proc Natl Acad Sci U S A. 1998, 95(26):15351-15355*). **c**, A strategy for screening enantioselectivity of ketoreductases. It is noteworthy that NADPH can be oxidized by the commonly used horseradish peroxidase (HRP) (*Sandro et al., Eur J Biochem. 1991, 201(2):507-513.*). To solve this problem, evolved HRP or other peroxidase species that are inactive toward NADPH would need to be developed.

Figure R6. A biosensor-based strategy that employs enantiomer-responsive transcriptional factors (a) and riboswitches (b) for the high throughput screening of enzymatic enantioselectivity.

We have added discussion points about the versatility of DMDS system in the discussion of the revised manuscript (page 15-16, line 348-354). These illustrations are also shown in the revised Supplementary Information (Fig. S14, S15).

Point 6: About the references: I think the first 6 references (on improving enantioselectivity by directed evolution) are a bit old. I understand you want to show some original references, but it would be better to add one or more recent ones as well, like: C.G. Acevedo-Rocha et al. 2010 (doi: 10.1016/j.jbiotec.2014.04.009), Otten et al. 2010 (doi: 10.1016/j.tibtech.2009.10.001), Reetz et al. 2010 (doi: 10.1351/PAC-CON-09-09-16), Denard et al. 2015 (doi: 10.1016/j.cbpa.2014.12.036), Xiao et al. 2015 (doi: 10.1021/ie503060a), Autour and Ryckelynck 2017 (doi: 10.3390/mi8040128), Reetz 2017 (doi: 10.1007/978-3-319-50413-1_3), Longwell et al. 2017 (doi: 10.1016/j.copbio.2017.05.012).

Response: We appreciate the reviewer's helpful suggestions. We have updated the references list as the reviewer suggested.

Point 7: Figure S5 is a bit strange. The synthesis of substrate 1 is in panel B, while the synthesis of substrates 2 and 3 are in panel A. Maybe this can be turned around for clarification?

In figures S6, S7 and S8 I had some problems in understanding the differences between the red and green bars. Maybe it can be added to the legend that red is activity (left axis) and green is enantioselectivity (right axis).

Response: We appreciate the reviewer's helpful suggestions. We have moved the synthesis of substrate **1** to panel A and moved the synthesis of substrates **2** and **3** to panel B. We have also added descriptions that *the red bar represents activity (left axis) and the green bar represents enantioselectivity (right axis)* to the legends of Figure S6, S7, and S8.

Point 8: Minor things

There are some typing mistakes in the following lines, including the omission of many articles, which should be changed: 11, 16, 23 87, 167, 179, 188, 205, 206, 207, 208, 211, 212, 213. Typing mistakes in the SuppInfo are on lines: 62, 92, 143.

In line 185 it says: ..higher than 99%. Mutant 6A8 is indeed >99%, but mutant 4D11 is 97%, so this should be: ..higher than 97%.

In line 162 and 163, the authors mention mutant 7A5, but I think this should be 7F9 (if this is correct, than fig. 2 is wrong). Also check Table S1 and S2 on this

issue.

Reference 3 and 6 are (almost) duplicates. Ref. 6 is correct, ref. 3 has the wrong year. Maybe the authors wanted to put this reference on position 3: Reetz et al. 2010 (doi: 10.1351/PAC-CON-09-09-16)?

Response: We apologize for the mistakes and greatly appreciate the care the reviewer took in reviewing our text. We have now checked the whole text carefully, and all similar mistakes have been corrected in both the revised manuscript and Supplementary Information.

Reviewer #3:

This paper describes a new screening protocol for directed evolution. This is a very important area because the most time consuming and prohibitive part of improving enzymes is the screening of the millions of mutants that are generated.

Overall I find the manuscript describe some very interesting science; however, it isn't clear to me how this revolutionizes the directed evolution process. This process seems very enzyme-substrate specific. How would this process work for a ketoreductase reaction, etc? In particular, I would expect most enzymes would not take a substrate that has a large dye attached to it due to active site size vs substrate size. As a result, the way it is written, I do not feel like this manuscript rises to the proper level to be accepted in this journal.

Response: We thank the reviewer for the helpful discussion. In general, a high throughput screening system consists of two subsystems: one is the detection subsystem that converts the PRODUCT of the enzymatic reaction into a detectable signal, and the other is a subsystem that reads the signal and enables isolation of the desired mutants accordingly. For instance, in a regular 96-well plate based screen, the design of chromogenic and fluorogenic SUBSTRATES are used as the detection subsystem to convert enzymatic activities into detectable absorbance or fluorescence changes, while 96-well plates themselves and a plate reader are employed to read the signal and obtain the desired mutants. In this paper, we mainly focus on the second type of subsystem, which is an ultrahigh throughput DMDS system that can detect two fluorescence signals simultaneously. Note that no modification of an enzymatic substrate is required

for any such screening system; it is perfectly normal and typical to produce the readout signal for the product after completion of any enzymatic reaction.

To this end, we have included several novel design elements into the new microfluidic detection/sorting device that serves as the core of the DMDS platform. For instance, we use optical fibers to achieve low-background detection limits and use carbon-PDMS microelectrodes for low-voltage sorting. We further developed two working modes for the DMDS platform, namely the *cooperative* mode and *biased* mode, to minimize false positives (a topic that has received considerable clarification in our revised text). To demonstrate the efficiency of the DMDS system, we conducted a series of screens for esterase for the chiral resolution of an important class of drugs, profens. The DMDS system overcomes the limitation of regular single-fluorescence FADS systems, which can only screen enzymatic activities because (in essence) they are single channel output devices. DMDS thus paves the way for the screening for enantioselectivity, substrate specificity, chemoselectivity, etc., in an ultrahigh-throughput manner. Besides, ever more complicated enzymatic properties can be screened through the DMDS system through further hardware or chemical technologies, for example by fusing target enzyme with a fluorescence protein, the enzymatic activity and expression level can be detected simultaneously, which allows the directly measuring of specific activity (for more in this vein, we direct the reviewer to our response for reviewer to comment Point 5 and to our discussion in the revised manuscript). Also, the DMDS system can in theory be used to detect the activities of more than one key enzyme in a metabolic pathway, which would be much more informative than typical detection of an end product. Above all, we believe that the DMDS system can broaden the application of droplet-based ultrahigh throughput screening.

Of course, the full exploitation of the potential of the DMDS system will require the corresponding deployment and/or development of subsystem 1 components. As we replied to Reviewer 2 (Point 5), this field has made great progress in recent years. The corresponding fluorogenic assays have been developed for all six major EC categories of enzymes (*Reymond et al., Chem Commun (Camb). 2009, (1):34-46. Goddard et al., Trends Biotechnol. 2004, 22(7):363-70.*), with a large number of commercial substrates are available. Especially the recently appeared new fluorescence detection strategies such as cascade enzyme-coupling detection and new biosensors based on transcription factors or riboswitches, obviate the dependence of large fluorophore modified non-natural substrates. These strategies can significantly expand the influence of FADS screening systems in general and especially our dual-channel system.

Comments:

This article seems to be written out of order. I was massively confused for the

first three pages. It seems like the narrative should start out at line 106 (this is where I started to get a better idea of their work and what was done)... and it seemed to be written more clearly (authors changed?) Similarly, they use the terms cooperative and biased mode and then don't define those terms until line 80. I suggest the authors step back and rethink how they want to explain this work. Perhaps, when better described, I would then better appreciate the impact that they feel this work will have on the improvement of the field of biocatalysis and enzyme evolution.

Response: Thank you for pointing this out; this comment has greatly guided our revision of the manuscript. Beyond implementing the suggested changes, we have more-or-less comprehensively redrafted the abstract, introduction, discussion, and methods sections. We have also substantively revised the results section as directed by the reviewers. We would also like to note that the entire revised text has been comprehensively edited for grammar and usage by a native-English-speaking professional editor with a PhD in biochemistry.

Side note: I collaborated with a computational chemist who did similar MD simulations (using autodock, etc) and many reviewers hated this work and said that this type of simulations weren't accurate enough to be published? It is outside of my field so I will leave it up to the editor to find a reviewer to comment on that portion.

Response: MD simulations are the core method employed in a large number of chemistry and biochemistry research disciplines, and are a widely-adopted means for predicting and gaining understanding about enzymatic mechanisms (large number of papers published annually). In this paper, we employed AUTODOCK4.2 for molecular docking between enzymes and substrates. We have now added simulation data performed using the Amber program: the enzyme-substrate structures were stable for as long as 20 ns during all the simulation process, indicating very high stability. We are firmly of the view that these results are reliable and provide meaningful insight into the mechanism of enzymatic enantioselectivity in our experimental system.

Reviewers' comments:

Reviewer #1 (Remarks to the Author):

The manuscript has been comprehensively redrafted. I agree it is now more readable and the logic behind the experiments is easier to follow. Beside this structural improvement, I still believe that the technological innovation, from a microfluidic point of view, is over claimed. It sounds more like the authors used existing FADS technology implemented with two substrates, which indeed allows for enantioselectivity screening (which is the real novelty of this work). The manuscript should not emphasize the innovation in terms of microfluidics. The authors claim that they have included several new designs elements into the new microfluidic detection/sorting device to improve already published FADS performance:

- i) use of optical fibers for fluorescence excitation, which has already been reported (Cole et al. 2015, doi:10.1039/C5LC00333D);
- ii) low voltage sorting (100 Vpp), which has already been described with same sorting frequency (1 300 Hz) (Romero et al. 2015, doi: 10.1073/pnas.1422285112).
- iii) sorting frequency of 1 400 Hz, which is better than the FADS first system described by Baret et al. (300 Hz, for similar false positives rates). Many other improved FADS designs have been described after the pioneer work of Baret et al. in 2009. Indeed, accurate FADS have been described with sorting frequency up to 30 kHz, with more than 99% accuracy (Sciambi et al. 2015, doi: 10.1039/C4LC01194E).

I'm also not convinced by the authors answer concerning my main comment (point 1). I still regret the lack of experimental evidence for the beneficial use of the cooperative mode in comparison with single substrate FADS. I do know the statement from Frances Arnold ("you get what you screen for") and I can understand that you can find examples in the literature where false evolutionary routes emerged from the use of a single substrate in specific cases (Aharoni et al. 2006). Nevertheless, as mentioned, it is a case-specific phenomenon, and if the authors want to emphasize on the fact that this cooperative mode is indeed beneficial in their evolutionary context, it has to be based on experimental evidence.

As a result, I do not feel that the manuscript raises the proper level for publication in Nature Communications in terms of microfluidic technological innovation.

Beside this, the results obtained in terms of directed evolution and enzyme engineering are impressive. This is however rather outside of my field for me to judge on their real innovation and impact level. I leave up this judgment to proper reviewer. The relevance of the computational part is outside my field of expertise.

Reviewer #2 (Remarks to the Author):

I want to compliment the authors on improving their manuscript. It was much easier to read this time, and I was happy to see all of my comments implemented in the revised manuscript. Together with the remarks of the other referees, I think the manuscript is now almost ready for publication in Nature Communications.

I have one comment on the addition of the discussion on the versatility of the DMDS system to the Supplementary Information (SI). I really enjoyed reading this extra part and I think this is crucial for the paper. This manuscript does describe a new system, so an outlook on future usage should be part of the main text and not of the SI. I don't know if there is a page limit to this manuscript

and therefor the authors decided to add this new part in the SI. If so I would prefer the conclusions to be shortened as this is a bit lengthy, and add this discussion there instead. But I guess this is up to the editor and authors to decide, and I will be OK with either outcome.

There are however some new typing mistakes and one missing reference.

In the abbreviations: ee Enantiomeric excess (there was an 'n' too many, and it is singular, not plural)

p2.called "dual-channel microfluidic droplet screening (DMDS) -> there are no closing "

legend Fig. 2 Use of DMDS-based directed evolution for the enhancement of the enantioselectivity of AEFST -> should be AFEST

Discussion, line 14: One the one hand should be On the one hand

Discussion, end of page 13. The incomplete information of enantiomeric purity readily undermines the majority of drugs are chiral molecules and that in most cases the potential side-effect risks curative effect of a chiral drug results mainly caused by from one of its enantiomers. This sentence is probably rewritten multiple times, which resulted in really bad English. This makes it really hard to understand what the authors mean, please rephrase.

Methods, libraries: reference 36 only contains the protocol for epPCR, not for DNA-shuffling or site-saturation mutagenesis. Please add references or the method.

p.15: The library was thusly condensed... thusly should be 'thus', or 'in this way'

p.16: The best mutant varient from this round... varient should be variant

Supplementary Information references:

19. González-García E et al...

29. De Sandro, V et al...

Reviewer #3 (Remarks to the Author):

The rewriting and editorial improvements has made a significant change to the "readability" of this manuscript.

Authors took significant time and did a great job answering the reviewers comments.

Response to Reviewers' comments:

As ever, we greatly appreciate the helpful comments from the reviewers on our manuscript entitled "**Efficient molecular evolution to generate enantioselective enzymes using a dual-channel microfluidic droplet screening platform**". Our point-by-point responses to the reviewers are presented below. Guided by these reviewer comments, we have made further corrections and improvements to our manuscript.

Our responses to the reviewers' comments

Reviewer #1:

Point 1: The manuscript has been comprehensively redrafted. I agree it is now more readable and the logic behind the experiments is easier to follow. Beside this structural improvement, I still believe that the technological innovation, from a microfluidic point of view, is over claimed. It sounds more like the authors used existing FADS technology implemented with two substrates, which indeed allows for enantioselectivity screening (which is the real novelty of this work). The manuscript should not emphasize the innovation in terms of microfluidics. The authors claim that they have included several new designs elements into the new microfluidic detection/sorting device to improve already published FADS performance:

- i) use of optical fibers for fluorescence excitation, which has already been reported (Cole et al. 2015, doi:10.1039/C5LC00333D);
- ii) low voltage sorting (100 Vpp), which has already been described with same sorting frequency (1 300 Hz) (Romero et al. 2015, doi: 10.1073/pnas.1422285112).
- iii) sorting frequency of 1 400 Hz, which is better than the FADS first system described by Baret et al. (300 Hz, for similar false positives rates). Many other improved FADS designs have been described after the pioneer work of Baret et al. in 2009. Indeed, accurate FADS have been described with sorting frequency up to 30 kHz, with more than 99% accuracy (Sciambi et al. 2015, doi: 10.1039/C4LC01194E).

Response: We agree with Reviewer #1 that one of the key innovations of this work is the capability of enantioselectivity screening by the combination of state-

of-the-art FADS technologies. We agree with the reviewer that we should focus on the enzyme engineering capacity and versatility of our DMDS system, rather than focusing on the particular performance parameters of the separate microfluidic components that comprise the overall system. Therefore, we have made several changes in the text to markedly tone down our claims regarding the novelty of the DMDS platform hardware.

Specifically, we have removed mention of

1) The key innovation underlying our new DMDS platform is its ability to evaluate two reaction channels simultaneously (page 2, line 60);

2) ...compared with previously reported excitation/emission setups that employ microscope optics (page 4, line 95);

3) Additional performance properties render the DMDS platform especially suitable for the efficient screening of enzymatic selectivity. For instance, whereas the initial technology that used FADS for screening, which was reported by Baret et al. (2009), could achieve a sorting rate of 300 droplet/sec at a voltage of 1000-1400 Vp-p with a false positive rate of ~1 in 10,000 (i.e., incorrect sorting of a non-fluorescent droplet),¹⁵ the new sorter design of our DC-FADS device achieves a sorting rate as high as 1400 droplets/sec at a much lower voltage (~100 Vp-p) and with a similar false positive rate of ~3 in 10,000. (page 4, line 105)

Point 2: I'm also not convinced by the authors answer concerning my main comment (point 1). I still regret the lack of experimental evidence for the beneficial use of the cooperative mode in comparison with single substrate FADS. I do know the statement from Frances Arnold ("you get what you screen for") and I can understand that you can find examples in the literature where false evolutionary routes emerged from the use of a single substrate in specific cases (Aharoni et al. 2006). Nevertheless, as mentioned, it is a case-specific phenomenon, and if the authors want to emphasize on the fact that this cooperative mode is indeed beneficial in their evolutionary context, it has to be based on experimental evidence.

As a result, I do not feel that the manuscript raises the proper level for publication in Nature Communications in terms of microfluidic technological innovation.

Beside this, the results obtained in terms of directed evolution and enzyme engineering are impressive. This is however rather outside of my field for me to

judge on their real innovation and impact level. I leave up this judgment to proper reviewer. The relevance of the computational part is outside my field of expertise.

Response: We thank the reviewer for this helpful discussion point and explanatory notes; we now realize that we were remiss in our first and second submissions in our assumptions about the potential for practically relevant rates of false positives in single-substrate FADS applications. We were (admittedly very clumsily) trying to make a theoretical, design-related argument, but actually used language which implied that this hypothetical concern was an ongoing and problematic issue experienced frequently in practice. This was obviously sub-optimal, and we regret not being more rigorous in our thinking and in our rhetorical choices in previous versions of the manuscript. To correct this problem in our scientific argument, we have now made changes in the text (page 4, line 111-121) about our claims regarding the *cooperative* mode's capability in eliminating false positives.

Meanwhile, we have emphasized the design flexibility of the two working modes of DMDS system in the screening of a range of enzymatic reactions by various fluorogenic strategies. Happily, this re-emphasis is very much in line with a specific request from Reviewer #2 (below) advising us to underscore the suitability of our DMDS platform for use with a variety of exciting assay/sensing technologies including FRET, enzyme-coupled assays, riboswitches, nano-sensors, etc.

Reviewer #2:

Point 1: I want to compliment the authors on improving their manuscript. It was much easier to read this time, and I was happy to see all of my comments implemented in the revised manuscript. Together with the remarks of the other referees, I think the manuscript is now almost ready for publication in Nature Communications.

I have one comment on the addition of the discussion on the versatility of the DMDS system to the Supplementary Information (SI). I really enjoyed reading this extra part and I think this is crucial for the paper. This manuscript does describe a new system, so an outlook on future usage should be part of the main text and not of the SI. I don't know if there is a page limit to this manuscript and therefor the authors decided to add this new part in the SI. If so I would prefer the conclusions to be shortened as this is a bit lengthy, and add this discussion there instead. But I guess this is up to the editor and authors to decide, and I

will be OK with either outcome.

Response: We thank the reviewer for this helpful suggestion. We have added emphasis about the potential versatility of our DMDS system to the *Discussion* section. Specifically, we have added the following content (page 13, line 339-350):

In recent years, a large variety of fluorogenic enzymatic assays have been developed for the high throughput screening of enzymes of all six of the major EC catalytic categories; these developments have greatly expanded the potential application scope for DMDS-based directed evolution screens.^{33, 34} Besides the fluorogenic surrogate substrates that we used in this study, other relevant strategies include for example fluorescence resonance energy transfer (FRET),²⁰ enzyme-coupled assays,³⁵ functional group selective reagents,³⁶ transcriptional factors,³⁷ riboswitches,³⁸ and nano-sensors,³⁹ most of which can be converted with very simple modifications into dual-fluorescence assays that are amenable for use with our DMDS system (the principles for using other screening chemistries and/or biomolecules and are illustrated in Figures S14 and S15). The DMDS system is a highly versatile screening platform that can be used for the directed evolution of complex enzyme properties such as regio-, chemo-, and enantio-selectivity.

Point 2: There are however some new typing mistakes and one missing reference.

In the abbreviations: ee Enantiomeric excess (there was an 'n' too many, and it is singular, not plural)

p2.called "dual-channel microfluidic droplet screening (DMDS) -> there are no closing "

legend Fig. 2 Use of DMDS-based directed evolution for the enhancement of the enantioselectivity of AEFST -> should be AFEST

Discussion, line 14: One the one hand should be On the one hand

Discussion, end of page 13. The incomplete information of enantiomeric purity readily undermines the majority of drugs are chiral molecules and that in most cases the potential side-effect risks curative effect of a chiral drug results mainly caused by from one of its enantiomers.

This sentence is probably rewritten multiple times, which resulted in really bad English. This makes it really hard to understand what the authors mean, please rephrase.

Methods, libraries: reference 36 only contains the protocol for epPCR, not for DNA-shuffling or site-saturation mutagenesis. Please add references or the method.

p.15: The library was thusly condensed... thusly should be 'thus', or 'in this way'

p.16: The best mutant varient from this round... varient should be variant

Supplementary Information references:

19. González-García E et al....

29. De Sandro, V et al...

Response: We apologize for these mistakes and greatly appreciate the considerable care that the reviewer took in reviewing our manuscript. We have corrected all of the mistakes highlighted by the reviewer and have re-checked the whole text carefully.

REVIEWERS' COMMENTS:

Reviewer #1 (Remarks to the Author):

The authors did a great job rewriting the manuscript. I'm satisfied with the revised version. I have no additional comments.

Response to Reviewers' comments:

Reviewer #1 (Remarks to the Author):

The authors did a great job rewriting the manuscript. I'm satisfied with the revised version. I have no additional comments.

Response: We greatly appreciate the helpful comments from the reviewers throughout the whole process in revising our manuscript entitled "**Efficient molecular evolution to generate enantioselective enzymes using a dual-channel microfluidic droplet screening platform**".